# Aggressive dominance can decrease behavioral complexity on subordinates through synchronization of locomotor activities

Rocio Soledad Alcala[1,6], Jorge Martin Caliva[1,2,6], Ana Georgina Flesia[3,4], Raul Hector Marin[1,2,5] & Jackelyn Melissa Kembro [1,2,5]*

Social environments are known to influence behavior. Moreover, within small social groups, dominant/subordinate relationships frequently emerge. Dominants can display aggressive behaviors towards subordinates and sustain priority access to resources. Herein, Japanese quail (*Coturnix japonica*) were used, given that they establish hierarchies through frequent aggressive interactions. We apply a combination of different mathematical tools to provide a precise quantification of the effect of social environments and the consequence of dominance at an individual level on the temporal dynamics of behavior. Main results show that subordinates performed locomotion dynamics with stronger long-range positive correlations in comparison to birds that receive few or no aggressions from conspecifics (more random dynamics). Dominant birds and their subordinates also showed a high level of synchronization in the locomotor pattern, likely emerging from the lack of environmental opportunities to engage in independent behavior. Findings suggest that dominance can potentially modulate behavioral dynamics through synchronization of locomotor activities.

[1] Universidad Nacional de Córdoba, Facultad de Ciencias Exactas, Físicas y Naturales, Instituto de Ciencia y Tecnología de los Alimentos (ICTA), Córdoba, Argentina. [2] Consejo Nacional de Investigaciones Científicas y Técnicas (CONICET), Instituto de Investigaciones Biológicas y Tecnológicas (IIByT, CONICET-UNC), Córdoba, Argentina. [3] Universidad Nacional de Córdoba, Facultad de Matemática, Astronomía y Física, Córdoba, Argentina. [4] Consejo Nacional de Investigaciones Científicas y Técnicas (CONICET), Centro de Investigaciones y Estudios de Matemática (CIEM, CONICET), Córdoba, Argentina. [5] Universidad Nacional de Córdoba, Facultad de Ciencias Exactas, Físicas y Naturales, Catedra de Química Biológica, Córdoba, Argentina. [6] These authors contributed equally: Rocio S. Alcala, J. Martin Caliva. *email: jkembro@unc.edu.ar

In human conduct it is well established that the social environment in which individuals live can influence behavior by enforcing patterns of social control, providing or not providing environmental opportunities to engage in particular behaviors, reducing or producing stress, and placing constraints on individual choice[1,2]. In the case of farm and zoo animals the social environment has also been widely studied given its strong implication in animal welfare[3]. Moreover, for social species, group housing (as opposed to individual housing) is considered essential to favor an enhanced behavioral repertoire. In particular, animals in social groups perform behaviors focused on regulating cohabitation dynamics. Within large groups, with unlimited food and water resources, it has been proposed that a tolerant social dynamic based on very low aggressions among flock members is observed[4–6]. On the other hand, within small groups, regardless of the abundance of resources, social hierarchies (i.e., individuals with a dominant or a subordinate status) are usually established through the frequent performance of aggressive behaviors[7,8] (see examples in pigs[9,10] and poultry[11,12]). Under these conditions, animals ranking high in the dominance hierarchy have precedence at the feed trough, waterers, nests, and other resources[13]. As a result, subordinate animals are not just the target of aggressions from dominant animals but are also denied access to resources, or must wait their turn[10,14–16]. Furthermore, subordinates can also show physiological mediators that are associated with a state of chronic stress[17–19]. In a worst case scenario, if aggression is not reduced or stopped and the animal cannot escape from aggressors, it can ultimately lead to death[20].

Dominant behavior is established over time and is dynamic in nature. In many species, this process begins when two or more unacquainted adult individuals are brought together, and fights usually occur until each individual has established a dominance-subordination relationship with each other[7]. Thus, dominance arises from the interaction of at least two individuals[21], where a more aggressive animal in a specific environmental/social context becomes dominant. Aggressions tend to decrease after a dominance/subordinate relationships is established[7,22]. Also, dominance structures can change over time, and are not clearly evidenced or present in all social groups. It should be noted that the effect of the social environment on the behavior of the members of a group is most likely beyond allowing or not the display of certain behaviors. On a more fundamental level, it could also act as a modulator of temporal behavioral patterns.

It is well known that behaviors, such as locomotion, do not occur randomly over time but rather show temporal organization such as ultradian rhythms (i.e. rhythms with periods <24 h[23,24]), long-term correlations (i.e. present behaviors depends on past behaviors, long term memory) and scale-invariant fractal dynamics[25–31]. In regard to this last property, fractal refers to a geometrical object that are composed of subunits (and sub-sub-units, etc.) that resemble the structure of the overall object (i.e. self-similarity), and for this reason are often referred to as scale-invariant since the same pattern is seen at different scales of observation. When applied to time series, fractal temporal patterns show irregular fluctuations across multiple time scales[25,27]. In the case of behavioral time series, fluctuations from the scale of seconds to hour have been observed[24]. Moreover, they show fluctuations that are complex (i.e. obey a scaling law indicating a fractal organization of their frequency components[32]). In this context, behavioral patterns represent an emergent property from underlying multi-scale physiological processes and their environmental interactions[21]. In specific, locomotor temporal patterns are particularly interesting given that they reflect both motivations to move (e.g., to feed, drink, or escape) and to remain immobile (e.g., when resting, fearful, threatened or hiding).

By studding temporal behavioral patterns, and not limiting our study exclusively to average values, it is possible to assess modifications in the dynamics of behavior, as well as evaluating the level of synchronization between animals' behavior[24]. Previous studies have shown the potential of this strategy to assess effects of social stressors. For example, in hens, an increase in animal housing density by the temporary addition of two animals (i.e. introduction of strangers) increased the complexity of locomotion (i.e. greater stochasticity)[33]. In wild primates, linear mixed-effects models suggest that low dominance status, infection, impaired health, reproductive activity, and ageing were all associated with reduction in locomotor complexity[34].

In the present study, we compare, under controlled laboratory settings, the temporal dynamics of behavior in small social groups of Japanese quail exhibiting divergent characteristics: (i) clear dominant/subordinate relationship or (ii) no apparent social ranking between group members (i.e., none or low levels of aggressive interactions; neutral relationship). Experimentally, the social groups corresponded to triads of two females and one male. Although this ratio 2:1 (female: male) is not typically used in poultry commercial production systems, in our laboratory setup it allowed assessment of female-female as well as female-male interactions, while avoiding well documented violent male-male aggressions. By continuously tracking each bird in these social group, we were able to quantitatively assess individual behavioral dynamics, and monitor possible behavioral synchronization between animals of the same group, using a combination of different mathematical tools. We show that the presence of a dominant bird leads to a decrease in the behavioral repertoire displayed by its subordinate but also affects the temporal organization of their behavior. Moreover, subordinates show a high level of synchronization in their locomotion patterns with their dominant counterparts. On the contrary, birds in non-aggressive neutral groups show non- correlated independent locomotor patterns. Thus synchronization between dominant/subordinate could reflect social stress and lack of environmental opportunities for the subordinate individual within the social environment.

## Results

**Assessment of variability in social group behavior.** We first evaluated behaviors performed by each member of the triad (Fig. 1a) immediately after being placed in the novel social environment (day 1) and 48 h later (day 3). Table 1 shows, in general, that both males and females on the first day spent less time eating and more time foraging in wood shavings and dust bathing in comparison to day 3. Also, the fractal analysis detrended fluctuation analysis (DFA, see Materials and Methods for details) on the locomotor time series showed that the dynamics of locomotion also varied over time in both females and males. Specifically, the self-similarity parameter ($\alpha$) which reflects the strength of long-range power-law correlations (i.e. degree of persistent behavior), and is a measure of self-similarity (fractality) in the temporal organization of a time series[35], showed lower values on the first day in comparison to day 3, indicating lower levels of long-range correlations (i.e. higher stochasticity) of the locomotor time series. No differences were found between days in percent of time spent performing ambulation, aggressive pecking and reproductive (grabs, mounts and cloacal contacts) behaviors. Interestingly, although the average number of pecks performed or received did not change between day 1 and 3 (Table 1), as flockmates became more familiar with each other, the proportion of individuals receiving the aggressive social interactions decreased. Specifically, during the first hour in the novel social group, 92% of the birds (33 out of 36) received pecks, while on the third day, a significantly ($P = 0.04$) lower proportion of birds received pecks (72%, 26 out of 36). Regarding the sequence

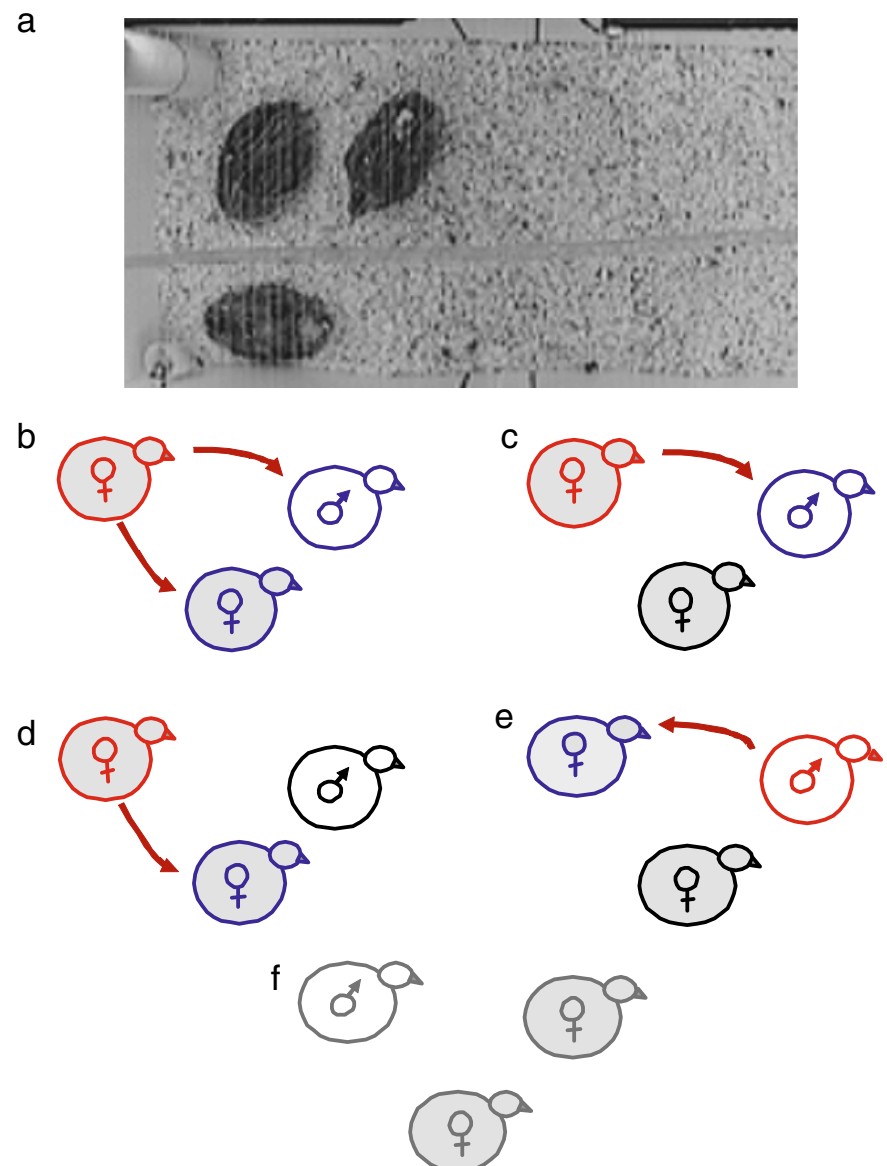

**Fig. 1 Social groups and aggressive social interactions within groups on day 3. a** The social group triad is observed as well as the feeder in top left corner and automatic nipple in bottom left corner of the apparatus. **b–f** Schematic representation of the direction of the aggressive social interactions (dark red arrow) present between individuals in the triad of two females (gray background) and one male (white background) 48 h after being placed in the novel social environment. Red indicates dominant birds (aggressive score ≥ 4.9), blue, subordinate birds (aggressive score ≤ −4.9) that receive pecks from dominant birds, and black those birds that are neither dominant nor subordinate (4.9 > aggressive score > −4.9). **f** represents neutral groups (i.e. the group did not include any dominant member).

of male reproductive behaviors, on day 1, 83% (11 out of 12) performed grabs, 75% (9 out of 12) mounts and 66% (8 out of 12) reached cloacal contacts. Similarly, on day 3 the proportion of males performing these behaviors was 75, 67 and 50%, respectively. Also, in general, females spent significantly ($P \leq 0.05$) more time performing pecks, and foraging, and tended to spend more time eating than males (Table 1). These last to variables could reflect higher energy requirements needed in females for daily egg laying and larger size[36] in comparison to males.

**Variability in social group dynamics**. Within social groups we evaluated social interactions on day 3, in particular pecking behavior performed and received by each bird. As expected, a large variability between birds were observed in the time spent performing pecks (from 0 to 18.2 s; Supplementary Tables 1 and 2)

toward an individual of their social triad. Thus, we focused on determining whether dominant/subordinate relationships could be detected in each social groups. Each individual was classified according to an aggressive score adapted from Hurst et al.[37] (see Materials and Methods for details) based on the time spent performing and receiving pecks as either dominant (i.e. pecked at conspecifics but received none or few pecks in return), subordinate (i.e. receive pecks from dominant birds) or neutral (i.e. performed and received none or few pecks). Three out of the seven birds classified as dominant were females that performed pecking behaviors towards both female and male conspecifics (Fig. 1b). There were also cases where dominant females pecked preferably at either the male ($n = 2$, Fig. 1c) or the female ($n = 1$, Fig. 1d) group member, and where the dominant male pecked at only one of the females (n = 1, Fig. 1e) (data set available in Supplementary

**Table 1 Time (in seconds[a]) spent performing behaviors during 1-h immediately after birds were placed in a novel social group (day 1), and 48 h later (day 3).**

| Variable | Day 1 | | Day 3 | | P-value | | |
| --- | --- | --- | --- | --- | --- | --- | --- |
| | Female (24) | Male (12) | Female (24) | Male (12) | Day | Sex | DxS |
| Eating (s) | 6.7 ± 4.2 | 3.4 ± 2.9 | 175.6 ± 34.3 | 71.9 ± 28.4 | 0.0002 | 0.08 | 0.91 |
| Foraging (s) | 286.7 ± 64.1 | 123.2 ± 28.3 | 67.3 ± 15.6 | 37.2 ± 20.3 | 0.0002 | 0.03 | 0.78 |
| Dust bathing (s) | 728.6 ± 125.1 | 659.4 ± 200.6 | 35.4 ± 18.0 | 15.0 ± 9.9 | 0.0001 | 0.36 | 0.46 |
| Pecks performed (s) | 3.9 ± 1.4 | 3.4 ± 1.3 | 5.4 ± 1.8 | 1.2 ± 0.9 | 0.37 | 0.05 | 0.10 |
| Pecks recieved (s) | 3.8 ± 1.0 | 3.8 ± 1.3 | 3.17 ± 1.11 | 5.59 ± 2.09 | 0.44 | 0.75 | 0.44 |
| $\alpha$-value | 0.74 ± 0.01 | 0.73 ± 0.01 | 0.84 ± 0.02 | 0.83 ± 0.03 | 0.0001 | 0.79 | 0.86 |
| Ambulation (%) | 20.9 ± 1.4 | 26.3 ± 3.0 | 22.1 ± 2.4 | 18.5 ± 3.2 | 0.13 | 0.78 | 0.08 |
| Grabs (s) | nd | 5.4 ± 3.6 | nd | 7.7 ± 5.1 | 0.11 | nd | nd |
| Mounts (s) | nd | 4.3 ± 1.3 | nd | 2.3 ± 1.3 | 0.15 | nd | nd |
| Cloacal contacts (s) | nd | 21.1 ± 9.2 | nd | 17.0 ± 9.8 | 0.01 | nd | nd |

Mean ± SEM. Sample size (*n*) is indicated next to treatment header in parenthesis
*nd* no data was obtained for the variable given that only the males perform those reproductive behaviors
[a]The behavior ambulation is exceptionally expressed as a percentage of time (%)

Tables 1 and 2). It sould be noted that although six out of the seven dominant birds were female, since twice as many females than males were present in each group, at a pobulation level, the proportion of dominance between sexes (25% of females vs. 9% of males) was not statistically different ($P = 0.38$). Five social groups presented all neutral birds (i.e. all triad members showed aggressive scores between −4.9 and 4.9,) and therefor were considered as neutral groups given that clear hierarchal ranks were not evident during the study (Fig. 1f).

**Behavioral complexity and social environment**. After determining on day 3 whether birds within the social triads were either a dominant, a subordinate, or a neutral individual, we then studied the dynamics of their locomotor time series. The DFA showed that all subordinate birds presented numerically higher $\alpha$-values ($\geq 0.93$) than dominant and neutral birds ($\alpha$-value $\leq 0.87$) (Figs. 2 and 3). In this context, the higher $\alpha$-value (closer to 1) indicates that activity fluctuations are characterized by strong long-range positive correlations, and thus are not dominated by random factors[28]. In contrast, the lower $\alpha$-value found in dominant and neutral birds (Fig. 2, dark gray in panel a and red line in b) indicate a more uncorrelated scale-invariance (more random activity fluctuations). In addition, these time series with lower $\alpha$-value have higher fractal dimensions[38] than the time series with a higher $\alpha$-value (Fig. 2, blue line in panel b).

When evaluating behavior on a population level, no differences were found in the total time spent ambulating ($F_{2,7} = 0.71$; $P = 0.67$, Fig. 4a) between dominant, subordinate or neutral females, however, significant differences were found in number of immobility events ($F_{2,7} = 9.01$; $P = 0.0017$, Fig. 4b) and $\alpha$-values ($F_{2,7} = 21.50$; $P \leq 0.0001$, Fig. 4c). Subordinate females showed both a lower number of immobility events (Fig. 4b) and a higher $\alpha$-value of locomotor time series (Fig. 4c) than their dominant counterparts, or than the birds within the neutral groups. Similar results were observed in males, however given that only one dominant male was detected, for statistical analysis the dominant male was excluded. As in females, while no differences were found between neutral and subordinate males in the total time spent ambulating ($F_{1,6} = 0.05$; $P = 0.38$, Fig. 4d), significant differences were found in their immobility events ($F_{1,6} = 10.24$; $P = 0.013$, Fig. 4e) and the $\alpha$-values ($F_{1,6} = 177$; $P \leq 0.0001$, Fig. 4f). No differences were found between the dominant and neutral individuals in the time spent ambulating, the number of immobility events, or the $\alpha$-value (Fig. 4).

Interestingly, during the 1 h of observation, only 10% of subordinate birds spent more than 60 s eating (Supplementary Fig. 2a), 45 s foraging (Supplementary Fig. 2b) or 0 s dust bathing (Supplementary Fig. 2c), in comparison with respectively the 83% ($P = 0.004$), 71% ($P = 0.02$) and 57% ($P = 0.1$) of dominant individuals, and 67% ($P = 0.01$), 47% ($P = 0.09$), and 40% ($P = 0.18$) of neutral individuals performing the behaviors over those thresholds (Supplementary Fig. 2a–c). No differences were found between the proportion of dominant and neutral individuals performing eating, foraging or dust bathing behaviors ($P > 0.38$). Thus, only the birds in subordinate positions appear with a reduced behavioral repertoire. Given that pecking behavior is highly and positively correlated with chasing ($R^2 > 0.89$, Supplementary Table 3), it should be considered that subordinate birds were also chased by the dominant ones.

The social groups with a dominant individual showed a significantly higher ($P = 0.03$) mean interindividual distance between group members than the groups with only neutral members ($48.5 \pm 2.1$ and $39.8 \pm 3.4$ cm, respectively), suggesting a lower social cohesion.

Multivariate analysis of behaviors exhibited by quails in social groups shows that the Principal Component 1 (PC1) explains 34.3% and 36.2% of the total variance in females and males (x-axis Fig. 5), respectively. The remaining percentages of eigenvalues for the other principal components were of 28.9% and 25.7% (PC2, y-axis Fig. 5), 16% and 22% (PC3), 12% and 10% (PC4), and 10% and 5% (PC5) for females and males, respectively. Analyzing the influence of each variable in the configuration of the first two components of PC (Supplementary Table 4 and Fig. 5), it is evident that a higher $\alpha$-value is associated with less time eating and foraging in females, and less time eating and performing reproductive behavior (i.e. grabs) in males.

**Synchronicity between dominant and subordinate birds**. Visual observation of video recordings (Supplementary Movies 1 and 2) and time series (Fig. 2b) evidenced signatures of synchronization of locomotion between dominant and subordinate animals. To quantitatively assess synchronization between birds within each social group, Wavelet analyses were performed on actograms of locomotor behavior. An example of the analyses is shown in Fig. 6. The real part of the continuous wavelet transform (cwt, Fig. 6b) of each of the corresponding actograms (Fig. 6a) is observed. In panel c the Spearman's rank correlation coefficients ($r^2$) estimated between cwt values at each time scale are shown,

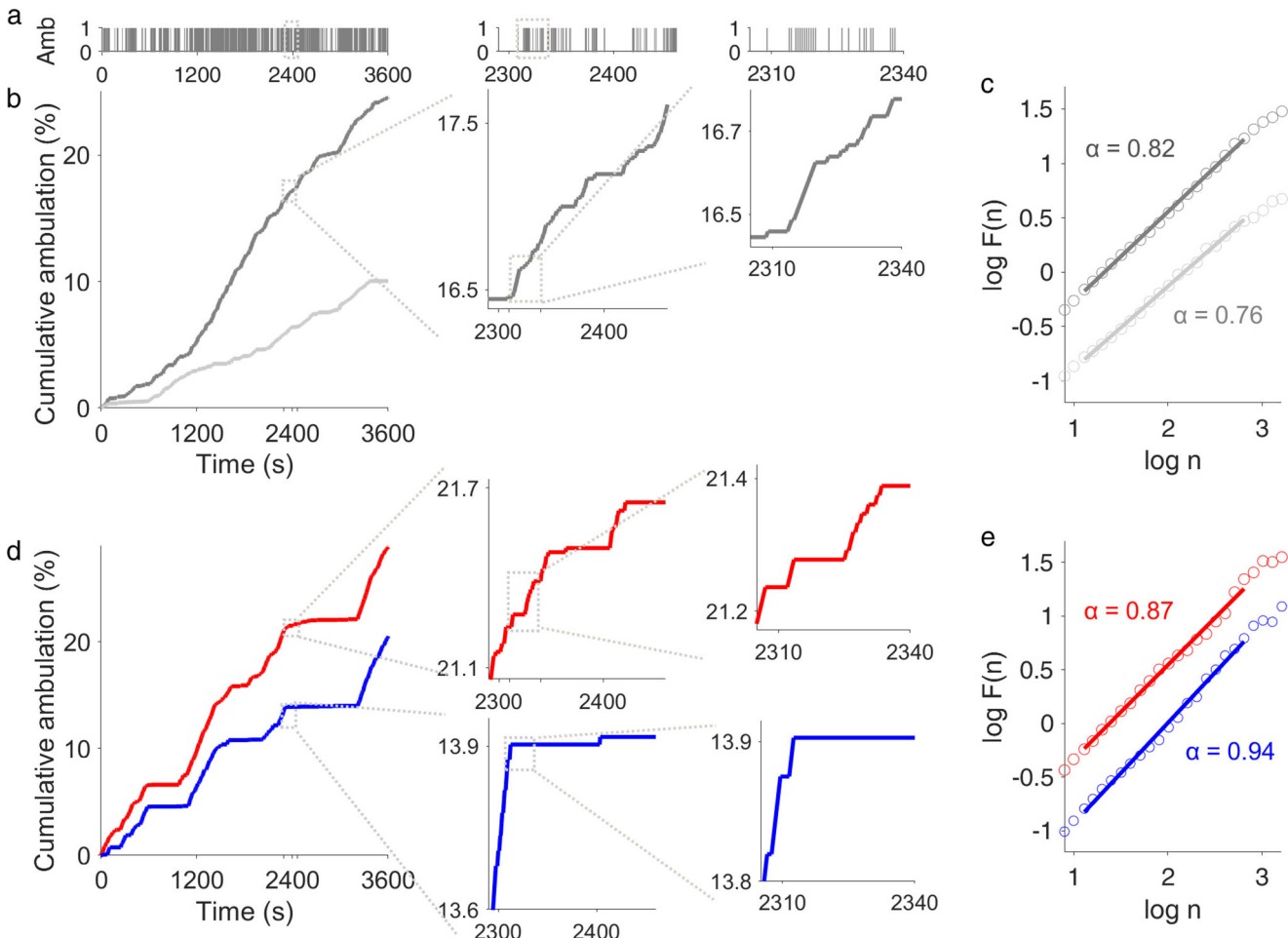

**Fig. 2 Graphical representation of scaling properties of locomotor time series of females within social groups 48 h after being placed in the novel social environment. a** Example of the same locomotor time series that is shown as a cumulative series in dark gray in b. Locomotion was monitored at 0.5 s interval ($x_i$); if the bird was ambulating, $x_i = 1$, and if immobile $x_i = 0$. Gray boxes mark the region amplified in insets. **b, d** Examples of cumulative locomotor time series of the two females within **b** a neutral social group (all group members were considered neutral) belonging to box 12 (Supplementary Table 1) or **d** a group with a dominant (red) and subordinate (blue) female (box 2, see Supplementary Table 1). Notice the similar pattern of activity and inactivity between dominant and subordinate birds. Gray boxes in panels b and d also represent the region of time series amplified in the inset. **c, e** Detrended Fluctuation Analysis (DFA) of the locomotor time series, corresponding to the same time series shown integrated in panels b and d respectively. Fluctuation functions were offset by 0.5 in order to improve visibility. Lines show the actual fitting region used ($n$, scales between 7.5 and 322 s) and the numbers represent the self-similarity parameter ($\alpha$-value) obtained for each of the locomotor time series. Note that the time series with the lowest $\alpha$-value such as those in **b**, present high level of switching between immobility and mobility events, thus shorter events (see also Fig. 5) as can be observed in insets. Higher $\alpha$-value (blue line in **e**) shows longer periods of continuous immobility or ambulation.

hence quantifying the level of synchronization of the activity pattern at the corresponding time scale. High-levels of synchronization ($r^2 > 0.6$) between birds are observed for wavelet time scales up to ~4 min. Evident from these plots is that a large positive value of the correlation coefficient at a given temporal scale indicates not only that the two animals compared present a peak in their locomotor activity at that scale, but also specifies their degree of synchronization with respect to the time at which activity appears. The association between correlation coefficient and synchronization between individuals can be visually verified by observing the overlapping in the real wavelet coefficients estimated from the locomotor time series of each animal. For example, in the top of panel d the values of both birds' real wavelet coefficients corresponding to the time scales with the highest correlation coefficient value (Fig. 6c at the 20 min) are depicted as a function of time. Here, a high level of synchronization between aggressive (red) and subordinate (blue) females is

observed. In the bottom of panel d, the values of both birds' real wavelet coefficients corresponding to the time scales with the lowest correlation coefficient value (Fig. 6c at the 5 s) are depicted as a function of time. In this case, the fine details of the actogram of each bird are highlighted and, as expected, a low level of synchronization is observed between animals.

At a population level, when the locomotor time series of dominant and subordinate individuals were compared, high levels ($r^2$ range 0.67–0.97) of ambulatory synchronization were observed across all wavelet time scales between 6 and 13 min (Supplementary Figs. 3a, 4, and 6a). Birds from neutral groups showed a higher relative dispersal given the larger diversity of values of correlation coefficients (Supplementary Figs. 3b, 5 and 6b) that range from −0.93 to 0.98. Also, for time scales up to 4 min the mean correlation coefficients were in general higher (mean $r^2 > 0.69$, Fig. 7a) and with lower relative dispersal (Fig. 7b) between dominant/subordinate quails compared to neutral/neutral

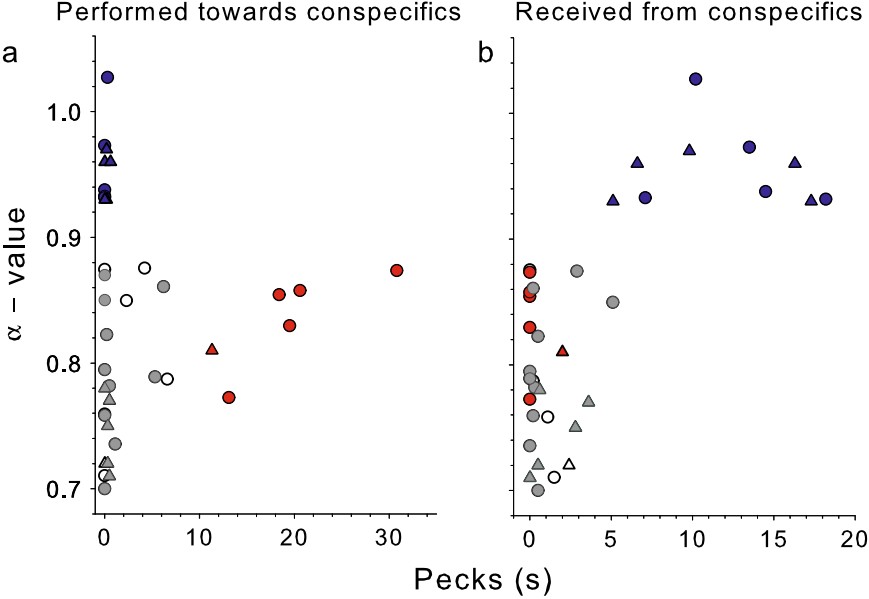

**Fig. 3 Subordinate birds show higher α-value in the dynamic of locomotion in comparison to dominant birds 48 h after being placed in the novel social environment.** The α-value estimated for locomotion time series of each bird in the social groups as a function of **a** the time spent performing aggressive pecks towards conspecifics and **b** the time receiving pecks from conspecifics during the 1-h period. It can be observed that dominant females (red circles) and the dominant male (red triangle) performed more than 6.5 s of pecks (**a**), received either 0 or <2 s of pecking (**b**), and showed α-values that range from 0.77 to 0.87. Subordinate female (blue circles) and males (blue triangles) quail performed either no pecks or ≤0.6 s (**a**), received pecks during more than 6.5 s (**e**) and never showed α-values lower than 0.93. Dark gray circles and triangles represent females and males that are in neutral groups were none of the members neither perform (**a**) nor receive more than 6.5 s of pecking (**b**). Open circles and open triangles represent males and females, respectively, that do not classify in any of these groups. The α-values were estimated using DFA3 (see Materials and Methods). Raw values are shown in Supplementary Tables 1 and 2. See Supplementary Fig. 1 for results from day 1.

individuals in neutral groups. For example, a tendency (Kruskal Wallis $P = 0.06$) towards lower levels of correlation were observed at 4.5 min between both types of social groups in females.

## Discussion

This study identified that birds that are subordinates show stronger long-range positive correlations in their temporal dynamics (higher α-values) of locomotion in comparison to their corresponding dominant counterparts. The increase in the strength of autocorrelation in the locomotor time series appears to be independent of the total time the birds spent ambulating but rather reflects a decrease in their behavioral repertoire. Quantifying the intuitive notion that the subordinate's behavior is subjugated by the dominant animal, we show that the activity pattern of the subordinate bird highly correlates on the scale of several minutes with that of it's aggressor, most likely reflecting behaviors associated with hiding (for example, staying immobile in a corner of the box, see Supplementary Movie 1) or being chased. On the contrary, in social groups where a clear dominant status is not present (neutral groups), a lower correlation between birds' activities were observed. To our knowledge, this is the first quantitative evidence of behavioral synchronization between dominant and subordinate animals within a social group, highlighting the relevance of adding tools of time series analysis, such as wavelets, in behavioral studies.

In poultry, the establishment of a peck order is expected to occur when unfamiliar birds are housed together in small groups[22]. Aggressions are also expected to be reduced as the flockmates became more familiar with each other[5]. In our study, although the average number of pecks performed or received did not change between day 1 and 3, as flockmates became more familiar with each other, the proportion of individuals receiving the aggressive social interactions decreased. Thus, day 3 could

represent an initial stage of establishment of dominant/subordinate relationships, suggesting continuity of enforcement of the pecking order with some birds perfoming a larger number of pecks towards conspecifics (dominant birds) while other birds are reducing at a minimum expression their aggressive behaviors (subordinates or neutral birds).

By assesement of the aggressive score, we observed by day 3 time that in 58% of the social groups clear dominant/subordinate relationships had been established, while the rest were non-aggressive neutral social groups. Consistently, social proximity, which is commonly associated with underlying sociality (motivation to be near conspecifics) and social cohesion[39], was also different between both group types. Groups with a dominant individual show lower social proximity (larger interindividual distance), than the groups with only neutral members, which is consistent with shyness and social withdrawal of subordinates, and decreased social cohesion. Quail classified as highly sociable in a social proximity test (i.e. density-related permanence test) also showed lower average distance between birds, and lower levels of aggression in comparison to those with low sociability[18]. Similarly, quail selected by their low andrenocortical response stayed closer together as chicks[39] and showed lower aggressiveness[39] as adults in comparison with those with high responsivenes. This diversity of types of aggressions can be considered a tradeoff between the long-term benefit of obtaining a dominant position to get priority access to environmental/social resources[40] and the associated energetic cost needed to obtain and maintain that position. How this cost/benefit relationship plays out depends on many factors including environmental characteristics as well as the animal's phylogeny, ontogeny, personality and prior life experiences.

In our study, the majority of neutral groups were composed of individuals that were a priori selected based on low fearfulness

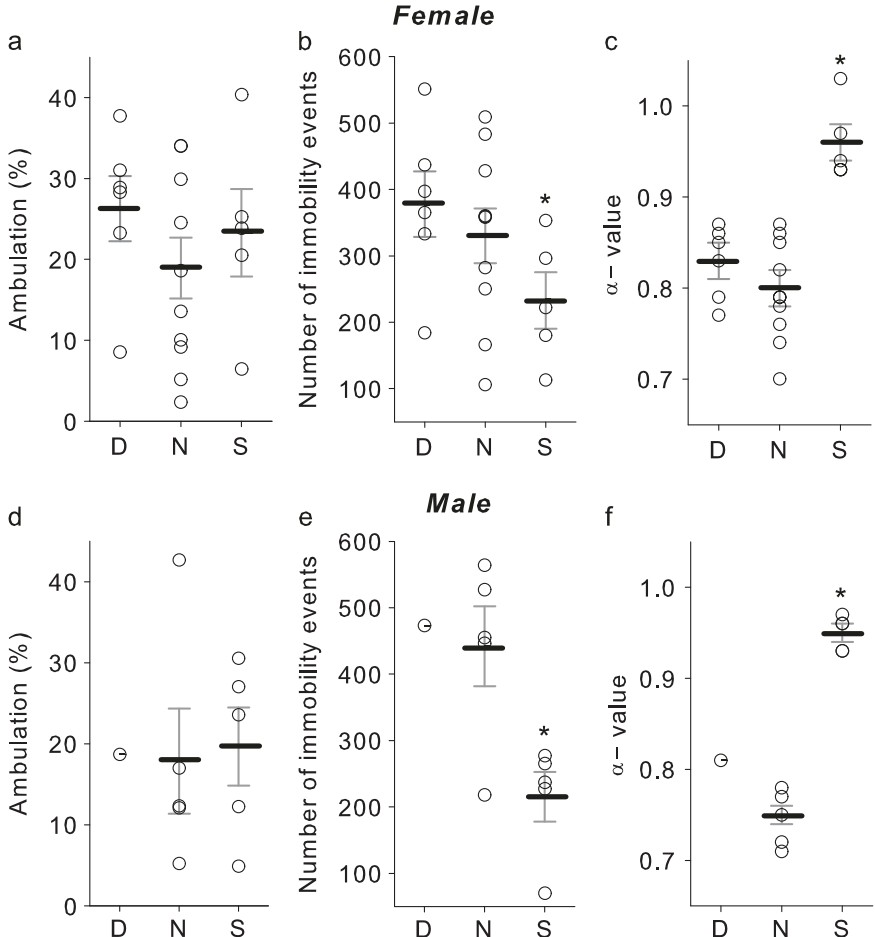

**Fig. 4 Higher levels of autocorrelation and less immobility events are observed in subordinate individuals in comparison to dominant and neutral birds 48 h after being placed in the novel social environment. a–c** show mean (dark black lines) ± SEM (gray lines) responses in females and **d–f** in males 48 h after being placed in the novel social environment. Raw data are shown in open circles. According to pecking behavior, birds were classified as Dominant (D), Subordinate (S) or belonging to a neutral (N) group. The dominant male was excluded from statistical analysis given there was only one representative from this group. In females, the sample size used was six dominants, 10 neutrals and five subordinates, while in males one dominant (not included in statistical analysis), five neutrals, and five subordinate. *differ at $P < 0.05$ from dominant and neutral individuals in females (**b**, **c**) and from neutrals in males (**e**, **f**).

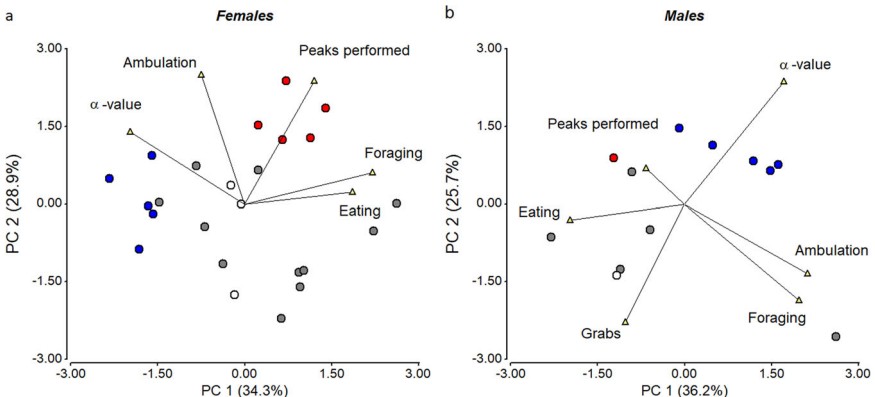

**Fig. 5 Exploration of behavioral variability of males and females within social groups at 48 h after being placed in the novel social environment.** Principal Component Analysis (PCA) Bi-plot graph. Circles represent (**a**) females or (**b**) males of each social groups. Full red circles ((●)), full blue circles ((●)), and full gray ((●)), indicate dominant and subordinate birds and birds in neutral groups, respectively. Open circles (o) indicate birds that do not fall into this classification. Triangles represent the variables used in the PCA, namely time spent ambulating, pecking at conspecifics, foraging, eating and performing grabs (males only), as well as the α-value estimated with DFA3 from locomotor time series. The percent of the eigenvalues of each PC are shown in brackets next to each component on the x- and y-axis.

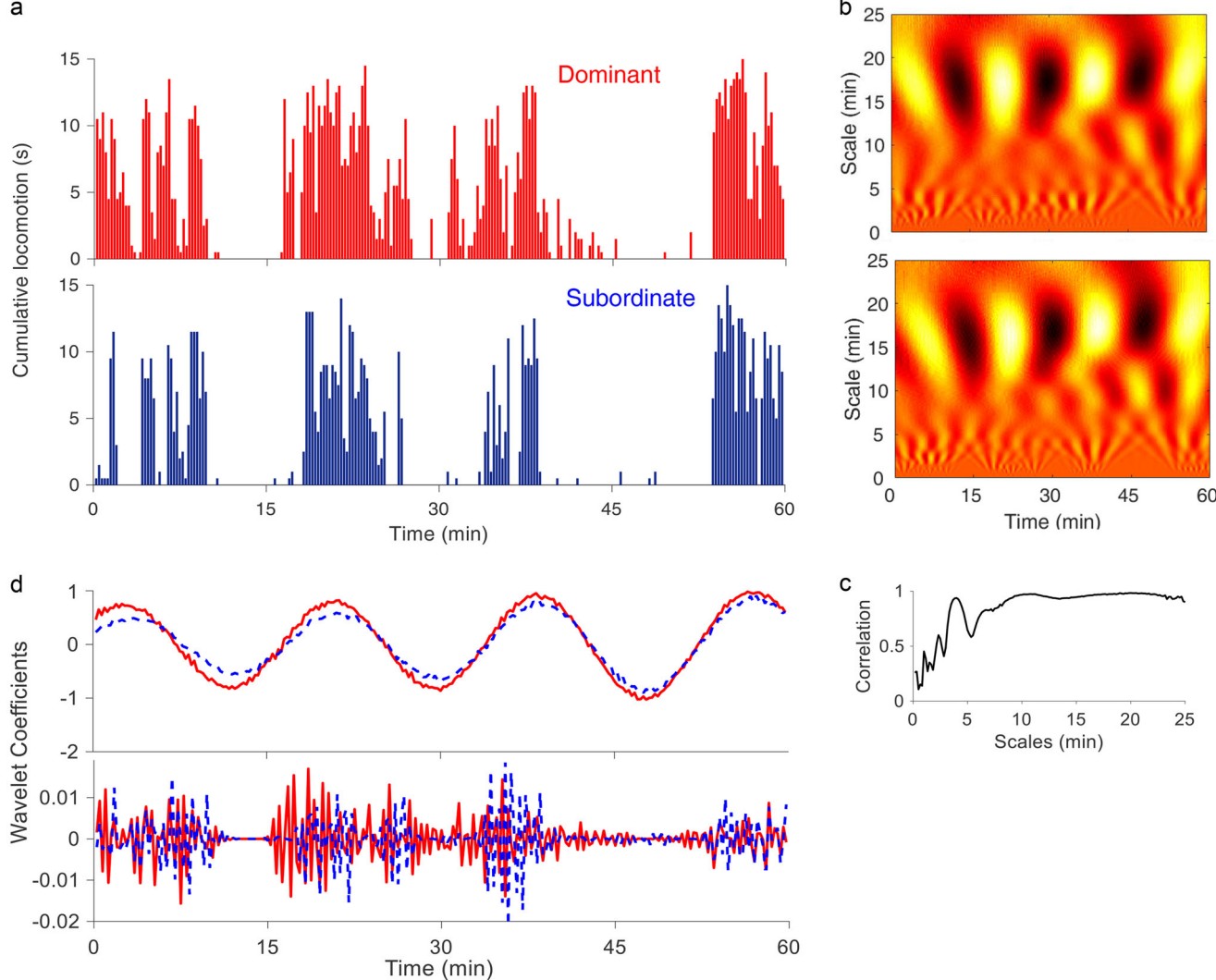

**Fig. 6 Synchronicity of locomotor dynamics between dominant and subordinate females within social 48 h after being placed in the novel social environment. a** Comparative actograms (15 s bins) of the same two locomotor time series represented in Fig. 2b, from dominant (top in red) and subordinate (bottom in blue) female Japanese quails within the same social group. **b** Plots of the real part of the wavelet coefficients estimated with a complex Morlet wavelet for the corresponding two time series shown in **a**. The x-axis represents time (60 min) and the y-axis indicates the scale of the wavelet used (from 5 s to 25 min). Yellow-white indicates higher values while black indicates minimal amplitude values. **c** Representation of the diagonal correlation coefficients obtained by comparing the real component of the wavelet coefficients for the same two animals at each time scale using the Spearman correlation coefficient. Note that these correlation coefficients denote the time interval where there is an effective correlation and the temporal scales at which the maximum value of the coefficients occurs. **d** Depicted is an example of the amplitude of the wavelet coefficients at 20 min (**d**, top panel) (maximal correlation), and 5 s (**d**, bottom panel) (minimal correlation) between the two time series shown in **a**.

and non-aggressiveness in behavioral tests (for details see Caliva et al.[41]). Thus, these neutral birds could present a specific coping style[42]/personality[42–44] which favor positive social interaction. Previous studies have shown that quail selected by their high andrenocortical response to restraint (i.e. propose to have a reactive personality[45]), are more fearful in a wide variety of tests[46–48] but, as stated previously, also are more aggressive in social groups[39], in comparison with those with low responsiveness (i.e. proactive personality[45]). In sheep, analysis of social behavior and the index of success of displacement has suggest the existence of at least 4 personality profiles (avoider, affiliative, aggressive, and pragmatic)[49]. In their study, sheep with the avoider and affiliative profiles do not use aggressive behaviors, but rather the nonagonistic behaviors (i.e. licking, grooming, sniffing) as their predominant social strategy. It is possible that in our

study all members of the neutral groups had profiles similar to the avoider and affiliative profiles, thus use nonagonistic behaviors as their predominant social strategy.

Although the hypothesis based on personality traits is plausible, other scenarios (or a combination of them) could also be considered to explain the outcome of neutral social environments in our experimental setup. First, because during development birds were reared in groups, it is expected that all tested birds were directly or indirectly involved in aggressive interactions and some of them could have therefore experienced social defeat. Animals that have lost fights often display a loser-effect (i.e. an individual that losing one encounter is likely to lose the next[50]) and tend to avoid the high cost of future aggressive confrontations[51]. However, it should be noted that during the first hour of exposure to the novel social environment test, pecking was

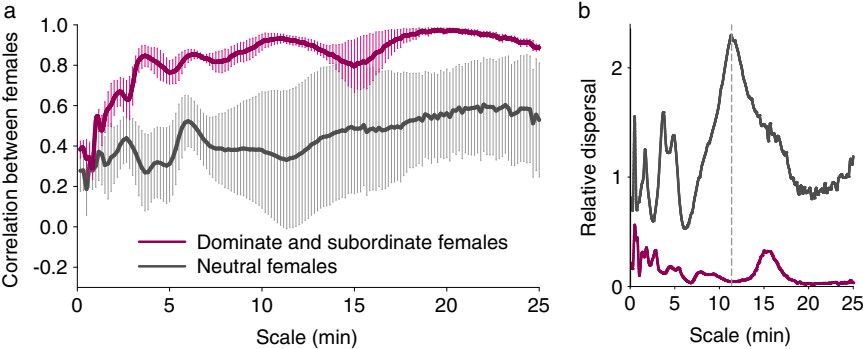

**Fig. 7 Synchronization in the locomotor activity between females within social groups, 48 h after being placed in the novel social environment.**
**a** Mean ± SEM of the pairwise comparisons of the real wavelet coefficients between the dominant and subordinate female (violet, $n = 4$ groups) or between two neutral females (dark gray, $n = 5$ groups) within the social group at each time scale was performed using the Spearman correlation coefficient. Supplementary Fig. 3 show the original four pairwise comparisons between dominant/subordinate females and the five pairwise comparisons between neutral females used for estimation of the mean values (see also Supplementary Figs. 4 and 5 for raw analysis, respectively). To reduce noise, Wavelet analysis was performed on actograms with 15 s bins. The same analysis performed in males is shown in Supplementary Fig. 6. **b** Relative dispersal (standard deviation/mean) of the curves shown in **a**. Dotted line represents a time scale in which the relative dispersal is about 50 times higher between neutral birds than between the dominant/subordinate females.

actually observed between individuals in some of those neutral social groups. Thus, a stable pecking order could have been quickly established in the neutral groups, and therefore directed pecking aimed at reinforcing the pecking order could have been deemed unnecessary. Secondly, since from an energetic standpoint, an animal should spend time and energy interacting with others only when this yields greater net benefits than an alternative behavioral strategy, such as ignoring others in the population and spending the time exploiting resources[52]. Considering that in our experimental setup food and water resources were provided ad libitum, a neutral group conformation would be energetically favorable for all members. Regardless of the underling motivations, in neutral environments all animals obtain the benefit of having access to resources (food and water) without the energetic cost of aggression. More importantly, all animals in the neutral group were able to show complex temporal behavioral dynamics (high fractal dimension, low α-value Fig. 3) and enriched behavioral repertoires (Fig. 4), and to our knowledge, particularly novel, a temporal-behavioral independence between group members.

In the social groups where dominant/subordinate relationships were established, the perceived benefit of a high status in the peck order in regard to priority of access to resources, is evidenced as a change in the time distribution assigned to different routine activities by dominant and subordinate birds. In rats, for example, dominant animals sleep more and low status individuals, especially females, spent more time moving around the enclosure and stretching up to the walls[37]. In our study, subordinate birds dedicate less time to feeding, dust bathing and foraging compared to Dominant or Neutral individuals. Interestingly, this change in the time allocated to those behaviors is also reflected in the temporal organization of the locomotion. Subordinate birds showed a lower degree of fractal complexity (i.e. higher α-value) compared to both the dominant counterparts and the birds in neutral groups. This is consistent with a previous study in wild primates where females with a high-ranking displayed greater complex behavior than low-ranking individuals both in foraging and locomotor behaviors, with mid-ranking individuals exhibiting an intermediate complexity. However, in their study, male locomotion sequences were generally less complex than those of females and the reverse was true among low-ranking individuals[34]. In our study, the expected behavioral differences between sexes were also observed. For example, males performed

reproductive behaviors, such as grabs mounts, and cloacal contacts, while female showed squats or avoidance behavior. However, neither clear differences in the temporal dynamics of locomotion were observed between females and males in general (Table 1), nor between the females and males that were ranked as subordinate individuals (females: 0.96 ± 0.02, males: 0.95 ± 0.02). Thus, we propose that the lower complexity found in the locomotor pattern of the subordinate birds, when compared to dominant and neutral birds, is associated with the social stress that these animals were subject to.

Social stress, as defined by Zayan[53], is elicited by behavioral actions from conspecifics leading to psychological stress, which has three components[53]: (i) negative emotional experiences (fear, suffering), (ii) a perceptive process by means of which familiar conspecifics are recognized as damaging (e.g. as despots), and (iii) cognitive processing through which animals anticipate the presence or actions of stressful conspecifics. In this context, once social hierarchy is established stress results not only from being constantly threatened and from having to inhibit one's own aggression in the near presence of a permanent dominant[54] but also, as a general rule, from having less control over partners and over environmental resources[53]. The association between behavioral complexity and social stress observed herein in quail, has also seen in pigs that when exposed to a mild chronic stressor (that included repeated aggressive social interactions with larger unfamiliar pigs) a reduced complexity in postural behavior in observed[55]. In sum, social stress resulting in a decrease in the animal's possibility to perform different tasks, can lead to decreased behavioral complexity. This contention is further supported by the observed synchronization between the subordinate and the dominant behavioral patterns.

The quantitatively assessment behavioral synchronization between individuals in the social groups was performed using Wavelet analysis. Interestingly for larger time scales, between 6 and 14 min, a high level ($r^2$ range 0.67–0.97) of locomotor synchronization was observed between dominant and subordinate birds in all groups presenting that relationship (Supplementary Figs. 3a and 6a). On the contrary, in social groups with all neutral animals, a larger diversity of levels of correlation (−0.93 and 0.98) was observed (Supplementary Figs. 3b and 6b). Synchronization in behavior has been reported in domestic fowl chicks[56] and in juvenile quail[57]. Lumineau et al.[57] showed social synchronization in the activity/rest cycle of a group of young Japanese quail.

However, to our knowledge this is the first time the level of synchronization between animals within a social group is quantified as a function of their social hierarchy. In contrast, socially neutral environments, apparently, provide more possibilities of behavioral independence between neighboring individuals. The lack of differences in locomotor dynamics ($\alpha$-value) observed between neutral and dominant animals further highlights the contention that what is lost in the subordinate counterparts is the lack of environmental opportunities to engage in independent behaviors.

Although behavioral motivations may be distinctly diferent between male and females, the impact of aggressions was not. Indeed, in both males and females, our study shows that the presence of a dominant aggressor within a group leads to an increase of the strength in long-range autocorrelations of locomotor dynamics and to a more limited behavioral repertoire in subordinates. In this context, the pattern of locomotion appears as an emergent property of the system as a whole encompassing the behavioral repertoire (e.g., hiding immobile, being chased, moving towards feeder or waterer, foraging). Thus, the high strength of correlations in behavioral of the subordinate emerges from the individual's dynamics within a social group where its pattern of locomotion shows a high level of synchronization with that of their dominant counterpart. Because the subordinate's behavior appears limited by dominant aggressor, it may reflect a lack of social space to perform independent locomotor behavior. Moreover, given that birds were restricted within the walls of the box, the subordinate animal cannot leave the social group and display a true retreat in order to place an end on aggressive interaction. Since the subordinate necessarily stays in the presence of dominant aggressor, a positive feedback loop can be established where aggressiveness towards the subordinate increases over time, further limiting the possibility of the subordinate to express normal complex behavioral patterns. As an extreme interpretation, this situation can even be paralleled with human bullying where a person is daily forced to continue interacting with an aggressive counterpart (bully), and thus cannot escape social stress (i.e. classrooms in school, a household).

In conclusion, our in-depth analysis of the effect of dominance on fractal behavioral complexity provides quantitative insight into its impact on the freedom to express behavior of the subordinate individual. We show that the capacity of an individual to display complex behavioral dynamics is dramatically affected by the presence of a dominant aggressor within their social environment. The dominant animal not only display highly complex behavioral patterns, similar to individuals in neutral (non-aggressive) social environments, but also their presence synchronizes activity patterns, at least within small groups. This synchronization is consistent with social stress and lack of environmental opportunities for the subordinate individual as suggested by their decreased behavioral repertoire and complexity. These findings may have broad implications for farm and zoo animal welfare by providing a framework for analysis of behavioral synchronization between dominant/subordinate individuals, as well as pave the way for exploring strategies to counteract or help in controlling aggressive behaviors. Moreover, our study illustrates a quantitative approach to studying correlations between the temporal dynamics of the individuals immerse in a specific social environment. This novel strategy can be readily applied in diverse fields such as behavioral ecology, sociobiology, neurobiology, and animal welfare.

## Methods

Extended detailed descriptions of all experimental procedures, publicly available raw data and video recordings obtained during this study are provided in the companion data descriptor article[41].

**Animals and husbandry**. The study was performed with Japanese quail (*Coturnix japonica*), a species widely used for studies covering neuroendocrine and social behavior studies[58,59]. The animals were bred according to standard laboratory protocols[60,61] and in accordance with the general rules of the National Research Council: guide for the care and use of laboratory animals[62]. The experimental protocol was approved by the Institutional Council for the Care of Laboratory Animals (CICUAL, Comite Institucional de Cuidado de Animales de Laboratorio) of the Instituto de Investigaciones Biologicas y Technologicas (IIByT, CONICET - Universidad Nacional de Córdoba). Mixed-sex Japanese quail hatchlings were randomly housed in groups of 50–60 in white wooden brood boxes measuring 90 × 80 × 60 cm (length × width × height, respectively) with a feeder along one wall, and 16 automatic nipple drinkers. A wire-mesh floor (1 cm grid) was raised 5 cm to allow the passage of excreta to the collection tray to facilitate cleaning and comfort of the animals, and a lid prevented the birds from escaping. Brooding temperature was 37.0 °C during the first week of life, with a weekly decline of 3.0 °C until room temperature (24–27 °C) was achieved. The first week of life all animals were raised under the same standard conditions. Quail were subjected to a daily cycle of 14 h light (300–320 lx): 10 h dark (long photoperiod; photostimulated) throughout the study, with the exception of Photocastrated stimulus birds (for the Social interaction test, see Supplementary material) that were submitted to a short photoperiod light cycle (06 h light: 18 h dark) beginning at 4 weeks of age until testing ended[63].

At 28 days of age, test animals were sexed by plumage coloration, marked with a wing band and randomly housed in pairs of one male and one female in cages of 20 × 40 × 20 cm (width × length × height, respectively).

Food and water were provided ad libitum. Both started and layer feeds were commercially obtained (20% of Crude Protein and 2900 kcal of Metabolizable Energy/kg diet). Feed contained corn, disabled soybean, wheat bran, soybean pellets, sunflower pellets, calcium, salt, vitamins, minerals and phosphate[64]. Although in this study feed consumption was not assessed, previous studies in our laboratory under similar conditions show a weekly feed intake of adults of 212 ± 2 g (~30 g daily)[64]. Birds were weighed at 28 days of age, and the weight of birds transferred to cages ranged between 100–150 g. Thereafter, weight was recorded weekly until 9 weeks of age and then at 92 days of age. At these same time points, male gonadal measurements showed complete development (Cloacal gonadal volume CGV > 1000 mm$^3$) in all males by 9 weeks of age. Female quail egg production was monitored throughout the study and all females reached peak egg production.

If an animal showed any indication of illness or escaped from their cage during rearing, they and their companion cagemate were excluded from the experiment. A total of 106 quail (53 females and 53 males) were subject to preselection tests (see following section). After preselection, 36 birds (24 females and 12 males) were studied in 12 novel social groups (see section Social Group testing). Within the social groups, females were classified (see Classification of birds based on social hierarchy section bellow and Supplementary Table 1) as 6 dominants, 10 neutrals, 5 subordinates and 3 non-determined. Males (Supplementary Table 2) were classified as 1 dominant, 5 neutrals, 5 subordinate and 1 non-determined.

**Preselection of quail**. Animals were preselected based on a combination of four behavioral tests (see Supplementary Methods, section Behavioral tests and Caliva et al.[41]) taking into consideration that quail that are more fearful also have been shown to be more aggressive[18,39]. Data is publicly available on figshare[65]. The order in which birds were evaluated in each of the preselection tests was randomized.

Half of the 12 social groups evaluated (see below for details) had birds of type A that were fearful in both the Tonic immobility[66] and the Partial mechanical restraint[47] tests. These fearful males also were aggressive in the tests of Social interaction[62] and/or impacts on the welfare of their cagemate[51,83]. It should be noted that these last two tests only applied to males since, as expected, predominately only males show aggressiveness during those tests situations.

The other half of the groups had birds with females and males that were not fearful in the Tonic immobility test, and males that tested non-aggressive in both the Social interaction test and in their home cage

**Social group testing**. Novel social groups (two females: one male) of animals (156–171-days-old) were housed in a white wooden apparatus measuring 80 × 40 × 40 cm (width × length × height, respectively) with wood-shavings on the floor. A feeder and an automatic nipple drinker were positioned in opposite corners of the apparatus (Fig. 1, left and right bottom corner of box in the photograph, respectively). Nylon monofilament line was extended over the top of the boxes with a 1 cm separation in order to prevent the birds from escaping without interfering with their visualization. A video camera was suspended 1.5 m above the box. Since only four social groups could be tested simultaneously, the setup was repeated three consecutive times. For convenience, boxes in which each social group were placed were numbered from 1 to 12. Boxes 1–4 were tested simultaneously first, 5–8 s and 9–12 last. The order of testing of each social group was randomized.

We used IdTracker[67] in MATLAB R2017a to register ambulation at 0.5 s intervals during a 1 h period (7200 time intervals) immediately after being placed in the test apparatus between 9 a.m. and 10 a.m., and 48 h after testing began. Behavioral data was recorded in the form of a time series of mutually exclusive

**Table 2 Definition of behavioral variables recorded in social groups using ANY-MAZE©.**

| Variable | Definition |
| --- | --- |
| Peck | When one bird raises its head and vigorously pecks the other bird's body. |
| Grabs | When a bird catches (grabs) with their beak the neck or head region of the other bird. |
| Mount | While performing a grab, the bird approaches the other bird from behind, and places both feet on the dorsal surface of its torso, stepping over the other birds' tail (adapted from[84]). |
| Cloacal contact | During mounting, the bird lifts his tail and tilts his pelvis underneath the other bird and briefly presses its cloaca against the other bird (adapted from[84]). |
| Threats | One bird raises its head and neck rapidly, moves forward and backward vigorously in the direction of the opponent without making physical contact (adapted from[84]). |
| Chase | A bird runs after another that is escaping (adapted from[85]). |
| Foraging | Pecking at the ground or actively moving litter with beak. |
| Feeding | Peaking at food in the feeding trough. |
| Dust bathing | Vertical wing shakes in a lying position[86]. |

states. At any given time, if the bird was moving a number one was recorded or a zero if immobile.

Time series of non-ambulation behaviors were obtained through visual observation of video recordings using as an interface ANY-MAZE@ to register behavior. For each bird, when the specific behavior was performed the corresponding key was pressed until the bird finished performing the behavior, thus a binary time series, $x_i$, sampled at up to two data points per second was constructed for each behavior. Where, $x_i = 0$ indicates that the animal is not performing the behavior, while $x_i = 1$ indicates that the animal is performing the behavior. From the behavioral time series both frequency and durations of behaviors as defined in Table 2 were estimated. Since both frequency and durations of behaviors are highly correlated, and that duration has the advantage of being a continuous variable. Durations, defined as time spent performing behavior, were used in analysis.

We used the ANY-MAZE© to register behaviors as described in Caliva et al.[63] during a 1-h period between 9 a.m. and 10 a.m. after 3 days of habituation to the novel setting[41], these time series are publicly available on figshare[68]. All data analysis and technical validation was performed by one observer. This observer was blinded regarding the prior history of the animals allocated in each group.

IdTracker[67] in MATLAB R2017a was used to register ambulation at 0.5 s intervals ($x_i$) during the same 1 h period that behaviors were recorded. The ambulatory time series (7200 time intervals) of each bird was obtained by assigning a number one ($x_i = 1$) if during the interval the bird was ambulating (i.e. moved more than 1 cm in any direction), or a zero ($x_i = 0$) if immobile. Locomotor time series are publicly available at figshare[69].

**Classification of birds based on social hierarchy**. As stated previously, we adapted the method proposed by Hurst et al.[37] to allocate quail to social ranks within their groups. For each dyad in the group, the agonistic score was estimated as the difference between the time spent pecking and the time receiving pecks from a given conspecific. Positive values represent animals that performed more pecks than received, while the inverse is true for negative values. The animals with the highest 20% of agonistic scores (≥4.9 s) were considered dominant, and thus the value of −4.9 was used as the counterpart to classify subordinate birds. Raw data is shown in Supplementary Tables 1 and 2.

For each dyad in the group, the aggressive score[37] was estimated as the difference between the time spent pecking and the time receiving pecks from a given conspecific. The top 20% (7/36) of the birds with the highest aggressive score towards one or both conspecifics showed positive scores above 4.9 s. In general, they performed more than 6 s of pecks and received from both conspecifics no pecks or a sum <2 s, and therefore were considered to show a dominant position in the pecking order within their group (Supplementary Tables 1 and 2). The counterparts, those birds that frequently received pecks without pecking in returns showed negative aggressive scores of less than, or equal to −4.9 s, were considered subordinates (Supplementary Tables 1 and 2).

Birds were monitored remotely through the camera-computer setup at least on 3 time points throughout the day. Testing was interrupted if signs of physical injuries were apparent or considered that withstanding behavior could lead to physical injuries. Noteworthy, the test originally was planned for a 5-day period, but was interrupted on the afternoon of the third day given the observed aggressions, in particular within group 9.

**Data analysis**. For each bird, percentage of time spent ambulating was estimated as: $Amb_\% = (\Sigma x_i/N) \times 100$, where $N$ is the total number of intervals (7200 time intervals/h). Similarly, actograms were computed for each bird by estimating the percent of time ambulating in 15 s bins ($N = 240$).

Detrended fluctuation analysis (DFA): the method utilized herein to determine scale-invariance and to evaluate the presence and extent of long-range correlations in the animal locomotor activity, was introduced by Peng et al.[70] and is described in detail elsewhere[24,71]. Briefly, DFA estimates the self-similarity parameter $\alpha$ that measures the autocorrelation structure of the time series. If $\alpha = 0.5$, the series is uncorrelated (i.e., random) or has short-range correlations (i.e., the correlations decay exponentially), whereas $0.5 < \alpha < 1$ indicates long-range autocorrelation (i.e., correlation decays as a power-law), meaning that present depends on past behavior[72]. Also, $\alpha$ is inversely related to a typical fractal dimension, so in this case, the value increases with increasing regularity (or decreasing complexity) in the time series.

The presence of nonstationarities in the signal[73], such as those associated with polynomial and sinusoidal trends[35], as well as the coarse-graining method[74] used to obtain the locomotor time series, can lead to crossovers in the scaling curve. Thus, the potential presence of crossovers were systematically studied for detrending orders 1 through 5[72]. A DFA of third order (DFA-3) was the lowest detrending order that eliminated trends in all series and therefore it was applied to all series for estimating $\alpha$. In addition, the optimal range of scales n[35] between 7.5 and 322 s (see linear fits in in Fig. 2c, e), was determined using the following criteria: stable values of local slopes, maximum coefficient of variation and minimum sum of squared residuals[24,71,75].

Herein, DFA calculations were performed with a customized script ran on MATLAB R2017a. DFA code is publicly available at the Physionet website (http://www.physionet.org/physiotools/dfa/) and the Matlab script used herein is also publically available[76].

Wavelet analysis: This analytical approach allows a signal to be decomposed in a flexible manner, providing simultaneously information on the presence of periodic behavior and its time localization[77–79]. Herein, we used the complex *cwt* Morlet wavelet in the first windowed Fourier transform to extract time-dependent features. When using a complex waveform, its transform is also complex, thus, the *cwt* coefficients can be represented by their real and imaginary parts, or amplitude and phase angle. Data analysis in this article was done using the wavelet toolbox of MATLAB R2017a, in particular the continuous wavelet transform function, *cwt*. We used the complex Morlet wavelet, *cmor1–1.5*, with scales that ranged from 1 to 150 corresponding to periods of 0.16 to 25 min. For convenience, scales were transformed into frequencies using the *scales2freq* function of MATLAB. In order to reduce noise, wavelet analysis was performed on actograms with 15 s bins. Script is publically available[80,81]. For the synchronicity analysis, we followed Pering et al.[79] using the real part of the *cwt* (*Re(cwt)*). For additional information see Supplementary information of Guzman et al.[24].

**Statistics and reproducibility**. Sample size was estimated assuming: (1) Dominant, subordinate, and neutral type animals would be attained. (2) Although birds in each group were preselected and allocated in type A and type B groups, the social dynamics of these new groups would not necessarily coincide with the expected behavior given their group type. (3) Based on Caliva et al.[63], for the expression of aggressive pecks, a standard variance of 3.3 was considered, a minimum difference between means of 4.5 seconds of aggressive pecks, a two-sided significance of 0.05 and a potency of analysis >80%. For DFA, a standard variance of 0.003 and a minimum significant difference of 0.15 was also considered[24,48,61,82]. From these estimations, a minimum sample size of 4 group types (with dominant/subordinate individuals or with only neutral birds) would be required, thus 12 groups were evaluated. Effectively, four groups with dominant/subordinate females and five neutral groups were detected with the methodology used. None of the groups nor animals were considered outliers. Normality was tested using a Shapiro-Wilks test. All statistical analyses were performed in R version 3.4.0, using the user-friendly interface InfoStat version 2017. A $P$-value $\leq 0.05$ was considered for statistical significance.

We used Generalized Linear Mixed Models (GLMM) to study mean differences between day 1 and day 3 in behavior in males and females. Day of testing and sex were considered as fixed effects, while animal ID, repetition of experimental setup

and social group (i.e. box) were used as random factors. Most variables presented a gamma distribution with the exception of ambulation, and $\alpha$-value which showed normal distributions.

GLMM were also used to evaluate the effect of social hierarchy (dominant, neutral or subordinate) on time spent ambulating, number of mobility events and $\alpha$-values. Repetition of experimental setup and social group (i.e. box) were used as random factors. These variables presented normal distribution. Given sex differences in motivation and behavioral repertoire females and males were evaluated separately. In females, the sample size used was 6 dominants, 5 subordinates and 10 neutrals, while in males 1 dominant (not included in statistical analysis), 5 subordinate and 5 neutrals. It should be taken into account that sexes were not able to analyzed together do to unbalanced design, given that only one dominant male was detected. LSD post hoc analysis was used to evaluate differences between social hierarchy groups.

To test differences in the proportions of birds that perform given behaviors, a two-tailed difference in proportion test was used.

Pearson correlation analyses were performed between all variables studied. Multivariate statistic, Principal Component Analysis (PCA), was performed to explore and describe general data variability, using the following explanatory variables: time spent ambulating, pecking at conspecifics, foraging, eating and performing grabs (males only), as well as the $\alpha$-value estimated with DFA3 from ambulatory time series. Variables selection took into consideration the correlation coefficients between variables, avoiding using variables that showed moderate or high levels of correlation between them ($r^2 > 0{,}65$).

A Kruskal Wallis test was used to compare the level of correlation at the 4.5 min time scale between the females of groups with dominant/subordinate birds with groups with only neutral.

**Reporting summary**. Further information on research design is available in the Nature Research Reporting Summary linked to this article.

## Data availability

The behavioral measurements, individual locomotor time series, and original video files are available in figshare with the identifiers https://doi.org/10.6084/m9.figshare.7117679.v1, https://doi.org/10.6084/m9.figshare.7117631.v1 and https://doi.org/10.6084/m9.figshare.c.4424327[41,68,69]. The behavioral tests used to preselect quail is also publicly available on figshare with the identifier https://doi.org/10.6084/m9.figshare.7122926.v1[65]. All mentioned data is described in Caliva et al.[41].

## Code availability

Computer codes used are described in Guzman et al.[80] and publicly available on Figshare (DFA[76] and wavelets[81]) with identifiers https://doi.org/10.6084/m9.figshare.1514975 and https://doi.org/10.6084/m9.figshare.1514976, respectively.

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

## Acknowledgements

We thank our colleague Dr. Diego A. Guzman who provided insight and expertize that greatly assisted the research, and Dr. Miguel A. Aon for comments and edits that profoundly helped to improve the paper. This research was financially supported by Fondo para la Investigación Científica y Tecnológica (FONCyT) grant N° PICT-2016-0282, Consejo Nacional para Investigaciones Científicas y Técnicas (CONICET), and Secretaría de Ciencia y Técnica (SeCyT), Universidad Nacional de Córdoba, Argentina. AGF, JMK and RHM are career members of CONICET. JMC has a PhD scholarship from the later institution.

## Author contributions

R.A., J.M.C., R.H.M., A.G.F., and J.M.K. conceived and designed research strategy, interpreted results, edited, and revised paper; R.A., J.M.C., and J.M.K. performed experiments; A.G.F., J.M.K. performed time series analysis; J.M.K. drafted the paper and wrote Supplementary Information.

## Competing interests

The authors declare no competing interests.
