## [Peer Review File · Communications Biology]

Reviewers' comments:

Reviewer #1 (Remarks to the Author):

The manuscript presents an investigation of how social interactions of individuals who are members of a social group affect the behavior of individual members of the group, depending on their social position compared to other group members – dominant, neutral or subordinate. The authors study small social groups of Japanese quail, and rise a novel hypothesis that social interactions among group members would be reflected in the various types of behavior displayed by the individuals as well as in the temporal patterns and organization embedded in their ambulatory activity. Modern concepts and approaches derived from fractal theory and stochastic process are utilized to provide quantitative assessment of the behavioral dynamics of individual group members, and to identify differences in dynamic characteristics based on the role and status of individuals in the social group. Further, through sophisticated wavelet-based analysis, the authors provide first evidence of coordinated behavior and synchronized ambulation patterns between group individuals with different social roles. Remarkably, the authors demonstrate how behavioral types and dynamical patterns involve in time as a function of increased familiarity among individuals. The validity of the reported observations is supported by several complimentary methods of analysis, measures of complexity and statistical tests. The manuscript is well written and presents an elegant study of how social interactions impact behavioral dynamics of the individual.

I recommend publication.

There are several technical points related to the details of analysis and interpretation of results that the authors should address before publication:

1. On page 4, line 68, the authors discuss temporal organization, long-term correlations and fractal dynamics and self-similarity in locomotion. This should be put in context of earlier works which have pioneered these concepts and demonstrated scale-invariant organization in animal and human locomotion – for example:

Ashkenazy Y, et. al. A stochastic model of human gait dynamics. *Physica A* 2002; 316 (1-4): 662-670.

Hu K, et. al. Non-random fluctuations and multi-scale dynamics regulation of human activity. *Physica A* 2004; 337(1-2): 307-318.

Hu K, et. al. The suprachiasmatic nucleus functions beyond circadian rhythm generation. *Neuroscience* 2007; 149(3): 508-517.

Ivanov PCh, et. al. Endogenous circadian rhythm in human motor activity uncoupled from circadian influences on cardiac dynamics. *Proc. Natl. Acad. Sci.* 2007; 104(52): 20702-20707.

2. On page 6, line 113-115, the authors connect the value of the self-similarity parameter α with higher level of fluctuation between ambulation and immobility events, and more complex dynamics with lower level of long-range correlations (lower α -values). This is not quite accurate. The DFA fluctuation function $F(n)$ depends on the size of fluctuations (standard deviation of the ambulation time series), however, the scaling exponent α does not depend on the size of fluctuations. Also, the α value reflects only the strength of long-range power-law correlations (i.e. degree of persistent behavior), and does not impart on the complexity of the dynamics – the DFA α is a linear measure of self-similarity (fractality) in the temporal organization of a time series. The authors should correct this in their interpretation.

Similar statements are made on page 9, line 169-171, and also on page 17, line 344-345, which should also be revised. Lower α value does not mean more complex fractal dynamics, but simply transition from long-range correlated scale-invariant behavior to a trivial uncorrelated (white noise) scale-invariance for $\alpha=0.5$.

3. In Table 1, page 6, one can notice that although there is large increase in eating time in both female and male in day 3, the ratio of time spent for eating between female and male during day 1 is similar to this ratio for day 3. Same can be observed for the ratio of foraging time between female and male – similar ratios for both day 1 and 3. Would that mean that there is certain constancy in social interaction between female and male independent of familiarity?

4. On page 9, Fig.2: Are the DFA α values obtained for the cumulative ambulation or for the ambulation time series of 0 and 1? This would give different values for the α exponents. An example of the ambulation time series and the corresponding DFA scaling function $F(n)$ would be instructive to be shown.

5. On page 8, figure 1, line 161, for birds that are neither dominant nor subordinate: perhaps the inequality should be with inversed sign " $4.9 > \text{aggressive score} > -4.9$ "?

6. On page 6, line 115, 'stochasticity' is misspelled.

On page, line 352, 'quantitative' is misspelled. Line 353, 'synchronization' is misspelled.

7. Data were recorded and analyzed for one hour between 9-10 am. Would the results differ if same analysis is performed in a different hour for the day, having in mind circadian effects?

8. Page 27 method section, regarding DFA analysis: effects of trends (including quasi-sinusoidal trends as shown in Fig.7) and other artifacts on the DFA results were first investigated in:

Hu K, et. al. Effect of trends on detrended fluctuation analysis. Physical Review E 2001; 64(1): 011114(19).

Chen Z, et. al. Effect of nonstationarities on detrended fluctuation analysis. Physical Review E 2002; 65(4): 041107(15).

Also, it has been demonstrated that binary time series (of 0 and 1) affect differently correlated ($\alpha > 0.5$) and anti-correlated ($\alpha < 0.5$) signals -- see:

Xu Y, et. al. Effects of coarse-graining on the scaling behavior of long-range correlated and anti-correlated signals. Physica A 2011; 390(23-24):4057-4072.

These potential effects on the results should be discussed with reference to prior works.

Reviewer #2 (Remarks to the Author):

This study investigated the effects of social environment on birds' behavioural complexity. The authors managed to quantitatively evaluate changes in the animals' behavioural repertoire and, particularly, in the temporal dynamics of their behaviour as hierarchy established. They found out that subordinates performed less complex temporal dynamics of locomotion, and that their locomotor pattern was synchronised with locomotion pattern of dominants. These are, as far as I know, novel information, which are relevant not only as basic research - e.g., studies on social behaviour, but also from an applied perspective - from the point of view of animal welfare.

The manuscript was well-written and, from my point of view, has no great flaws that could prevent it from being published. Regarding data analysis, the choice of GLMM was appropriate; however, evaluation of the use of the "Detrended Fluctuation" and "Wavelet" analysis is outside of the scope of my expertise. The authors' interpretation and conclusions are also adequate.

Following I list some points in which the manuscript deserves improvement, mostly for clarity:

1 - Table 1: the time unity is missing (I assumed the unity is "seconds", but this must be explicit in the legend);

- 2 - Line 133: "Diversity in the establishment of dominance relationships within social groups" is unclear; you must state clearly - diversity of what? Diversity in the process of dominance establishment? In the relationships?
- 3 - Lines 169-171: "...with a lower α -value (Fig. 2, both dark gray in "a" and red line in "b") represent a more complex fractal dynamic, a higher fractal dimensions and a "rougher" appearance than the time series with a higher α -value (Fig. 2, blue line in "b") which can be associated with faster switching between ambulation and immobility in the time series" - this part needs rephrasing because, as it is, the final part of the sentence seems to refer to the "higher α -value". Besides, I suggest changing the term "rougher" for a more technical one;
- 4 - Legends of figures 1 and 4: I suggest removing the explanation of the technique used to define dominants, subordinates and neutrals. On the other hand, information on the circumstance data was collected (after 48 hours of group-forming) is missing in these and other legends;
- 5 - Try to reduce the number of figures. I suggest removing figures 5 and 8;
- 6 - Lines 253-255: "The social groups with a dominant individual showed a significantly higher ($P = 0.03$) mean interindividual distance between group members than the groups with only neutral members" - This result was not discussed in the Discussion section;
- 7 - Legend of Figure 6 - Review for clarity;
- 8 - Lines 382-385: "Considering that in our experimental setup food and water resources were provided ad libitum, no actual immediate benefit is obtained through the establishment of dominance, thus a neutral group conformation would be energetically favorable for all members." This argument is not valid, since there were groups, kept under the same conditions, where hierarchy was established.
- 9 - Some paragraphs of the manuscript are too long; split them into two or three for clarity (e.g., lines 355-399; 400-450);
- 10 - Lines 510-511: State clearly how many birds made part of the experiment - how many dominants, how many subordinates, how many neutrals. This is important information for reproducing the experiment;
- 11 - Table 2 - Please, align text in the left-hand column cells with the first line of the corresponding cell in the right-hand column;
- 12 - Supplementary Tables 1 and 2 - Please, revise the Table legends; they inform the recorded interactions took place "between each female" (1) or "between each male" (2), but both Tables include data from interactions with males and females. Besides, the information included below the Tables need revising as well ("...pecking towards and time receiving pecks"; make it clear - towards who?);
- 13 - Supplementary Figure 1 - The legend and the title of the X axis contains the word "peaks", instead of "pecks";
- 14 - Supplementary Figures 3 and 4 - The figures have several parts, and no identification was made of each part;
- 15 - Supplementary material - Lines 129-130 - The last sentence of the paragraph is incomplete.

Reviewer #3 (Remarks to the Author):

The study entitled "Aggressive dominance can decrease behavioral complexity on subordinates through synchronization of locomotor activities" analyze, under the perspective of the influence of the social environment on individual behavior from the perspective of a complex system in birds, the temporal dynamics of behavior in small social groups of birds exhibiting divergent characteristics. The study evaluate the temporal dynamics of the behavior and spatial use of the individual within their social environment, by using mathematical tools from the field of study of complex systems: scale-based analysis and DFA. This approach permit to obtain a precise quantification of patterns that are often limited to average durations of a behavior or the use of traditional patterns, which are often limited to average durations of a behavior. By continuously tracking each bird authors were able to quantitatively assess individual behavioral dynamics, and monitor possible behavioral synchronization between animals. Authors hypothesized that the presence of a dominant bird has an effect not only on the type of behaviors displayed by its subordinate but also on the temporal organization of behavior. The study address questions about

the influence of the social environment on the individual behavior, highlighting if there are biological rhythms and if these rhythms emerge from social dynamics. These questions are aimed at deepening knowledge of social dynamics using powerful tools of analysis, and study the problem of aggression in farm birds that threatens animal welfare. It is therefore a basic study of social behavior using very innovative techniques with results that, although expected, can give arguments to respond to the management challenges of social groups in captivity and their impact on the final efficiency of the system. Therefore, I consider that this work should be published with the certainty that it will be very useful both for the applied ethology research groups and for the analysts and managers of the captive bird production systems.

Line 68. Please cite the following article in this context:

Maria, G. A., Escós, J., & Alados, C. L. (2004). Complexity of behavioural sequences and their relation to stress conditions in chickens (*Gallus gallus domesticus*): a non-invasive technique to evaluate animal welfare. *Applied Animal Behaviour Science*, 86(1-2), 93-104.

Line 68 to line 82. Why is it avoided the most representative social groups of social production groups (ie laying hens, free range chicken) where there are groups only of females or only of males that although it is not a "natural" social group is the most frequent in management of these production systems?

Line 103. Temporal organization of the group?

Line 134. Endpoint criteria provided in cases of extreme aggression?

Line 140-142 The definition of the group called neutral is not clear, it seems that these animals can eventually move to well-defined categories as dominant and subordinate?

Line 355-365 How do you think that subordinate birds can be affected their productive efficiency? ... Was growth and conversion of food measured? ... Is it likely that these birds that spend more time walking develop a strategy to obtain resources with an avoidant personality of the dominant ones and which are finally more efficient? ... Can the energy cost of the dominant ones be higher than the energy cost of developing a strategy of avoidance and a better use of resources?

Line 366-374 Maybe it is more appropriate to talk about different "personalities" or adaptive strategies or coping styles of individuals and maybe it is good to review the literature in this regard, I recommend citing the following work:

Miranda-de la Lama, G. C., Pascual-Alonso, M., Aguayo-Ulloa, L., Sepúlveda, W. S., Villarroel, M., & María, G. A. (2019). Social personality in sheep: Can social strategies predict individual differences in cognitive abilities, morphology features and reproductive success?. *Journal of Veterinary Behavior*.

Page 18. Density and lighting are two of the aspects that affect the social groups of bird production. In your case, the density is not inconvenient and the lighting follows a natural regime. Taking into account the high densities (ie broiler) used in production and the 24-hour light rhythm during the first and last stage of fattening, the results obtained here can be useful in intensive bird production systems. ?

Line 479. Explain the "broad implication for farm animal welfare"...

Line 485. Give more information about the feeding regime of the animals and performance traits (consumption, efficiency, diet etc.)

Density and lighting are two of the aspects that affect the social groups of bird production. In your case, the density is not inconvenient and the lighting follows a natural regime. Taking into account the high densities (ie broiler) used in production and the 24-hour light rhythm during the first and last stage of fattening, the results obtained here can be useful in intensive bird production systems. ?

Reviewers' comments:

Reviewer #1 (Remarks to the Author):

The manuscript presents an investigation of how social interactions of individuals who are members of a social group affect the behavior of individual members of the group, depending on their social position compared to other group members – dominant, neutral or subordinate. The authors study small social groups of Japanese quail, and rise a novel hypothesis that social interactions among group members would be reflected in the various types of behavior displayed by the individuals as well as in the temporal patterns and organization embedded in their ambulatory activity. Modern concepts and approaches derived from fractal theory and stochastic process are utilized to provide quantitative assessment of the behavioral dynamics of individual group members, and to identify differences in dynamic characteristics based on the role and status of individuals in the social group. Further, through sophisticated wavelet-based analysis, the authors provide first evidence of coordinated behavior and synchronized ambulation patterns between group individuals with different social roles. Remarkably, the authors demonstrate how behavioral types and dynamical patterns involve in time as a function of increased familiarity among individuals. The validity of the reported observations is supported by several complimentary methods of analysis, measures of complexity and statistical tests. The manuscript is well written and presents an elegant study of how social interactions impact behavioral dynamics of the individual.

I recommend publication.

Re: Thank you very much for your positive comments, detailed analysis of our work and constructive comments. We have addressed all the points mentioned, incorporated all recommended papers as references and modified Figure 2 in order to include the ambulation time series and the corresponding DFA scaling function $F(n)$.

There are several technical points related to the details of analysis and interpretation of results that the authors should address before publication:

1. On page 4, line 68, the authors discuss temporal organization, long-term correlations and fractal dynamics and self-similarity in locomotion. This should be put in context of earlier works which have pioneered these concepts and demonstrated scale-invariant organization in animal and human locomotion –

for

example:

Ashkenazy Y, et. al. A stochastic model of human gait dynamics. *Physica A* 2002; 316 (1-4): 662-670.

Hu K, et. al. Non-random fluctuations and multi-scale dynamics regulation of human activity. *Physica A* 2004; 337(1-2): 307-318.

Hu K, et. al. The suprachiasmatic nucleus functions beyond circadian rhythm generation. *Neuroscience* 2007; 149(3): 508-517.

Ivanov PCh, et. al. Endogenous circadian rhythm in human motor activity uncoupled from circadian influences on cardiac dynamics. *Proc. Natl. Acad. Sci.* 2007; 104(52): 20702-20707.

Re: We agree that these are very important papers and have now cited them accordingly.

2. On page 6, line 113-115, the authors connect the value of the self-similarity parameter α with higher level of fluctuation between ambulation and immobility events, and more complex dynamics with lower level of long-range correlations (lower α -values). This is not quite accurate. The DFA fluctuation function $F(n)$ depends on the size of fluctuations (standard deviation of the ambulation time series), however, the scaling exponent α does not depend on the size of fluctuations. Also, the α value reflects only the strength of long-range power-law correlations (i.e. degree of persistent behavior), and does not impart on the complexity of the dynamics – the DFA α is a linear measure of self-similarity (fractality) in the temporal organization of a time series. The authors should correct this in their interpretation.

Similar statements are made on page 9, line 169-171, and also on page 17, line 344-345, which should also be revised. Lower α value does not mean more complex fractal dynamics, but simply transition from long-range correlated scale-invariant behavior to a trivial uncorrelated (white noise) scale-invariance for $\alpha=0.5$.

Re: These statements have now been rewritten to improve accuracy on this very important concept throughout the manuscript. In particular, the phrase on page 9 now reads:

In this context, the higher α -value (closer to 1) indicates that activity fluctuations are characterized by strong long-range positive correlations, and thus are not dominated by random factors²⁸. In contrast, the lower α -value found in dominant and neutral birds (Fig. 2, dark gray in "a" and red line in "b") indicate a more uncorrelated scale-invariance (more random activity fluctuations). In addition, these time series with lower α -value, have a higher fractal dimensions³⁷ than the time series with a higher α -value (Fig. 2, blue line in "b").

3. In Table 1, page 6, pos one can notice that although there is large increase in eating time in both female and male in day 3, the ratio of time spent for eating between female and male during day 1 is similar to this ratio for day 3. Same can be observed for the ratio of foraging time between female and male – similar ratios for both day 1 and 3. Would that mean that there is certain constancy in social interaction between female and male independent of familiarity?

Re: Yes, seemingly females eat and forage approximately twice as much as males both on day 1 and 3. It is an interesting point, and we speculate that it is most likely associated with the high energetic requirements of egg production in females. It is important to note that in quail, as in other poultry species, females place roughly one egg per day along several months (Mills et al., 1997, Neurosci Behav 21: 261-281). Also, females are larger (heavier) (Mills et al., 1997, Neurosci Behav 21: 261-281) than males in this species. Unfortunately, our experimental setup does not allow us to accurately estimate individual feed consumption nor feed conversion in order to further explore this assumption. Nevertheless, a sentence informing the readers about quail male/female characteristic is now added in the results section as follows:

Also, in general, females spent significantly more time performing pecks, and foraging, and tended to spend more time eating than males (Table 1). These last two variables could reflect higher energy requirements needed in females for daily egg laying and larger size in comparison to males ³⁶.

4. On page 9, Fig.2: Are the DFA α values obtained for the cumulative ambulation or for the ambulation time series of 0 and 1? This would give different values for the α exponents. An example of the ambulation time series and the corresponding DFA scaling function $F(n)$ would be instructive to be shown.

Re: The cumulative ambulatory time series were used herein for visualization purposes only, given it is easier to compare these series than the original locomotor time series. As the reviewer is well aware of, in the first step of DFA the time series were integrated.

An example of original locomotor time series is now presented in panel "a". The DFA scaling function $F(n)$ have been to Figure 2c,e. Figure 2 is now presented as follows (changes in legend are underlined):

Fig. 2. Graphical representation of scaling properties of locomotor time series of females within social groups

a) Example of the same locomotor time series that is shown as a cumulative series in dark gray in “b”. Locomotion was monitored at 0.5 s interval (x_i); if the bird was ambulating, $x_i = 1$, and if immobile $x_i = 0$. Grey boxes mark the region amplified in insets. b,d) Examples of cumulative locomotor time series of the two females within b) a neutral social group (all group members were considered neutral) belonging to box 12 (Supplementary Table 1) or d) a group with a dominant (red) and subordinate (blue) female (box 2, see Supplementary Table 1). Notice the similar pattern of activity and inactivity between dominant and subordinate birds. Grey boxes in panels b and d also represent the region of time series amplified in the inset. c,e) Detrended Fluctuation Analysis (DFA) of the locomotor time series, corresponding to the same time series shown integrated in “b” and “d” respectively. Fluctuation functions were offset by 0.5 in order to improve visibility. Lines show the actual fitting region used (n , scales between 7.5 and 322 s) and the numbers represent the self-similarity parameter (α -value) obtained for each of the locomotor time series. Note that the time series with the lowest α -value such as those in panel b, present high level of switching between immobility and mobility events, thus shorter events (see also Fig. 5) as can be observed in insets. Higher α -value (blue line in panel e) shows longer periods of continuous immobility or ambulation.

5. On page 8, figure 1, line 161, for birds that are neither dominant nor subordinate: perhaps the inequality should be with inversed sign “4.9 > aggressive score > -4.9”?

Re: This typo was corrected.

6. On page 6, line 115, ‘stochasticity’ is misspelled.

On page, line 352, 'quantitative' is misspelled. Line 353, 'synchronization' is misspelled.

Re: These typos have been corrected.

7. Data were recorded and analyzed for one hour between 9-10 am. Would the results differ if same analysis is performed in a different hour for the day, having in mind circadian effects?

Re: They definitely could be different. We have previously documented both circadian as well as ultradian rhythms in locomotion (Guzman et al., 2017, Sci Rep. 7(1):684). Moreover, the time of day was specifically selected given that in the morning both locomotor and reproductive activity is maximum (see Delville et al., 1986, Horm Behav 20: 13-22). Although we have not explored afternoon and evening time points, we have explored between 10-11:30 on day 3 and similar results are evident. A larger study covering full day dynamics are definitely within our plans.

8. Page 27 method section, regarding DFA analysis: effects of trends (including quasi-sinusoidal trends as shown in Fig.7) and other artifacts on the DFA results were first investigated in:

Hu K, et. al. Effect of trends on detrended fluctuation analysis. Physical Review E 2001; 64(1): 011114(19).

Chen Z, et. al. Effect of nonstationarities on detrended fluctuation analysis. Physical Review E 2002; 65(4): 041107(15).

Also, it has been demonstrated that binary time series (of 0 and 1) affect differently correlated ($\alpha > 0.5$) and anti-correlated ($\alpha < 0.5$) signals -- see:

Xu Y, et. al. Effects of coarse-graining on the scaling behavior of long-range correlated and anti-correlated signals. Physica A 2011; 390(23-24):4057-4072.

These potential effects on the results should be discussed with reference to prior works.

Re: These papers are fundamental and were taken into account in the analysis process. Citations have now added to the Materials and Methods accordingly, as follows (changes are underlined).

The presence of nonstationarities in the signal ⁷², such as those associated with polynomial and sinusoidal trends³⁵, as well as the coarse-graining method ⁷³ used to obtain the locomotor time series, can lead to crossovers in the scaling curve. Thus, the potential presence of crossovers were systematically studied for detrending orders 1 through 5 ⁷¹. A DFA of third order (DFA-3) was the lowest detrending order that eliminated trends in all series and therefore it was applied to all series for estimating α .

In addition, the optimal range of scales n^{35} between 7.5 and 322 s (see linear fits in in Fig. 2c,e), was determined using the following criteria: stable values of local slopes, maximum coefficient of variation and minimum sum of squared residuals^{24,70,74}.

Once again, we really appreciate the comments received and if the Reviewer and or Editor think that further changes should be made, please do not hesitate to contact us again.

Reviewer #2 (Remarks to the Author):

This study investigated the effects of social environment on birds' behavioural complexity. The authors managed to quantitatively evaluate changes in the animals' behavioural repertoire and, particularly, in the temporal dynamics of their behaviour as hierarchy established. They found out that subordinates performed less complex temporal dynamics of locomotion, and that their locomotor pattern was synchronised with locomotion pattern of dominants. These are, as far as I know, novel information, which are relevant not only as basic research - e.g., studies on social behaviour, but also from an applied perspective - from the point of view of animal welfare.

The manuscript was well-written and, from my point of view, has no great flaws that could prevent it from being published. Regarding data analysis, the choice of GLMM was appropriate; however, evaluation of the use of the "Detrended Fluctuation" and "Wavelet" analysis is outside of the scope of my expertise. The authors' interpretation and conclusions are also adequate.

Re: We would like to thank the reviewer for the positive feedback. We greatly appreciate the remarks aimed at improving the organization and clarity of our work. We have addressed all comments in this revised version. If the Reviewer and /or Editor think that further changes need to be done, please do not hesitate to let us know.

Following I list some points in which the manuscript deserves improvement, mostly for clarity:

1 - Table 1: the time unity is missing (I assumed the unity is "seconds", but this must be explicit in the legend);

Re: The following statement has been added to the legend: "The time unit seconds (s) was used for all behavioral variables, except ambulation which is expressed as a percentage of time (%)".

2 - Line 133: "Diversity in the establishment of dominance relationships within social groups" is unclear; you must state clearly - diversity of what? Diversity in the process of dominance establishment? In the relationships?

Re: In order to improve clarity, the subtitle was modified to "Variability in social group dynamics".

3 - Lines 169-171: "...with a lower α -value (Fig. 2, both dark gray in "a" and red line in "b") represent a more complex fractal dynamic, a higher fractal dimensions and a "rougher" appearance than the time series with a higher α -value (Fig. 2, blue line in "b") which can be associated with faster switching between ambulation and immobility in the time series" - this part needs rephrasing because, as it is, the final part of the sentence seems to refer to the "higher α -value". Besides, I suggest changing the term "rougher" for a more technical one;

Re: These sentences have been completely rewritten in order to improve accuracy, and also considering reviewer #1 comments, it now reads:

In contrast, the lower α -value found in dominant and neutral birds (Fig. 2, dark gray in "a" and red line in "b") indicate a more uncorrelated scale-invariance (more random activity fluctuations). In addition, these time series with lower α -value, have a higher fractal dimensions³⁷ than the time series with a higher α -value (Fig. 2, blue line in "b").

4 - Legends of figures 1 and 4: I suggest removing the explanation of the technique used to define dominants, subordinates and neutrals. On the other hand, information on the circumstance data was collected (after 48 hours of group-forming) is missing in these and other legends;

Re: Figure legends 1 and 4 were rewritten accordingly (changes underlined), in the rest of the figures in the titles the phrase "48 hours after being placed in the novel social environment" was added.

Fig. 1. (...) b-f) Schematic representation of the direction of the aggressive social interactions (dark red arrow) present between individuals in the triad of 2 females (grey background) and 1 male (white background) 48 hours after being placed in the novel social environment. Red indicates dominant birds (aggressive score ≥ 4.9), blue, subordinate birds (aggressive score ≤ -4.9) that receive pecks from dominant birds, and black those birds that are neither dominant nor subordinate ($4.9 > \text{aggressive score} > -4.9$). Panel "f" represents neutral groups (i.e. the group did not include any dominant member).

Fig. 4. Higher levels of autocorrelation and less immobility events are observed in subordinate individuals in comparison to dominant and neutral birds 48 hours after being placed in the novel social environment.

Panels a, b and c show mean (\pm SEM) responses in females and panels d, e and f in males 48 hours after being placed in the novel social environment. According to pecking behavior, birds were classified as Dominant (D), Subordinate (S) or belonging to a neutral (N) group. (...)

5 - Try to reduce the number of figures. I suggest removing figures 5 and 8;

Re: Figure 5 has been removed and placed as Supplementary Figure 2. We would prefer to keep Figure 8 in the main text, given it shows the actual mean group values estimated (a highly relevant information). However, we do not want to preclude publication because of this required change and we are willing to move it to the Supplementary material if the Reviewer and/or Editor feels strongly that is necessary.

6 - Lines 253-255: "The social groups with a dominant individual showed a significantly higher ($P = 0.03$) mean interindividual distance between group members than the groups with only neutral members" - This result was not discussed in the Discussion section;

Re: A discussion about this result is now included as follows:

Consistently, social proximity, which is commonly associated with underlying sociality (motivation to be near conspecifics) and social cohesion³⁹, was also different in both group types. Groups with a dominant individual show lower social proximity (larger interindividual distance), than the groups with only neutral members, which is consistent with shyness and social withdrawal of subordinates.

7 - Legend of Figure 6 - Review for clarity;

Re: Figure legend has been reviewed, and now reads as follows (main changes are underlined):

Fig. 6. Exploration of behavioral variability of males and females within social groups at 48 hours after being placed in the novel social environment.

Principal Component Analysis (PCA) Bi-plot graph. Circles represent A) females or B) males of each social groups. Full red circles (●), full blue circles (●), and full gray (●), indicate dominant and subordinate birds and birds in neutral groups, respectively. Open circles (○) indicate birds that do not fall into this classification. Triangles represent the variables used in the PCA, namely time spent ambulating, pecking at conspecifics, foraging, eating and performing grabs (males only), as well as the α -value estimated with DFA3 from locomotor time series. The percent of the eigenvalues of each PC are shown in brackets next to each component on the x- and y-axis.

8 - Lines 382-385: "Considering that in our experimental setup food and water resources were provided ad libitum, no actual immediate benefit is obtained through the establishment of dominance, thus a neutral group conformation would be energetically favorable for all members." This argument is not valid, since there were groups, kept under the same conditions, where hierarchy was established.

Re: The phrase was rewritten to improve accuracy and the statement about "Considering that in our experimental setup food and water resources were

provided ad libitum, a neutral group conformation would be energetically favorable for all members."

9 - Some paragraphs of the manuscript are too long; split them into two or three for clarity (e.g., lines 355-399; 400-450);

Re: Each of these two paragraphs have now been broken down into 3 paragraphs (please see main manuscript).

10 - Lines 510-511: State clearly how many birds made part of the experiment - how many dominants, how many subordinates, how many neutrals. This is important information for reproducing the experiment;

Re: All detailed information about how many birds were dominant, subordinate or neutrals is now included in the Methods Section "Animals and husbandry:

11 - Table 2 - Please, align text in the left-hand column cells with the first line of the corresponding cell in the right-hand column;

Re: Alignment has now been changed.

12 - Supplementary Tables 1 and 2 - Please, revise the Table legends; they inform the recorded interactions took place "between each female" (1) or "between each male" (2), but both Tables include data from interactions with males and females. Besides, the information included below the Tables need revising as well ("...pecking towards and time receiving pecks"; make it clear - towards who?);

Re: Table legends now read:

Supplementary Table 1. Individual female dyadic pecking interactions with both male and female conspecifics in the social group.

Supplementary Table 2. Male dyadic pecking interactions with both female conspecifics in the social group.

13 - Supplementary Figure 1 - The legend and the title of the X axis contains the word "peaks", instead of "pecks";

Re: Typo has been corrected.

14 - Supplementary Figures 3 and 4 - The figures have several parts, and no identification was made of each part;

Re: These legends have been rewritten to clarify this aspect:

Supplementary Figure 3. The four panels represent each of the four social groups where clear dominate-subordinate relationships are observed. Within each panel, the pairwise

comparison of the real part of wavelet coefficients between the subordinate and the dominant female within each social group for the full range of time scale is shown.

Supplementary Figure 4. The five panels represent each of the five social group with all neutral members. Within each panel, the pairwise comparison of the real part of wavelet coefficients between the female within neutral social group for the full range of time scale is shown.

15 - Supplementary material - Lines 129-130 - The last sentence of the paragraph is incomplete.

Re: The sentence has been completed as follows: If during the interaction a quail received more than 5 consecutive aggressive pecks, showed a clear and continued escaping (retrieval) behavior, and/or showed any sign of physical injuries testing was finalized immediately.

Reviewer #3 (Remarks to the Author):

The study entitled “Aggressive dominance can decrease behavioral complexity on subordinates through synchronization of locomotor activities” analyze, under the perspective of the influence of the social environment on individual behavior from the perspective of a complex system in birds, the temporal dynamics of behavior in small social groups of birds exhibiting divergent characteristics. The study evaluate the temporal dynamics of the behavior and spatial use of the individual within their social environment, by using mathematical tools from the field of study of complex systems: scale-based analysis and DFA. This approach permits to obtain a precise quantification of patterns that are often limited to average durations of a behavior or the use of traditional patterns, which are often limited to average durations of a behavior. By continuously tracking each bird authors were able to quantitatively assess individual behavioral dynamics, and monitor possible behavioralsynchronization between animals. Authors hypothesized that the presence of a dominant bird has an effect not only on the type of behaviors displayed by its subordinate but also on the temporal organization of behavior. The study address questions about the influence of the social environment on the individual behavior, highlighting if there a biological rhythms and if this rhythms emerge from social dynamics. These questions are aimed at deepening knowledge of social dynamics using powerful tools of analysis, and study the problem of aggression in farm birds that threatens animal welfare. It is therefore a basic study of social behavior using very innovative techniques with results that, although expected, can give arguments to respond to the management challenges of social groups in captivity and their impact on the final efficiency of the system. Therefore, I consider that this work should be published with the certainty that it will be very useful both for the applied ethology research groups and for the analysts and managers of the captive bird production systems.

Re: Thank you very much for the detailed analysis of our work and the positive comments. We have now addressed all the points mentioned. Once again, if the Reviewer and /or Editor think that further changes need to be done, please do not hesitate to let us know.

Line 68. Please cite the following article in this context: Maria, G. A., Escós, J., & Alados, C. L. (2004). Complexity of behavioural sequences and their relation to stress conditions in chickens (*Gallus gallus domesticus*): a non-invasive technique to evaluate animal welfare. *Applied Animal Behaviour Science*, 86(1-2), 93-104.

Re: Yes, it is indeed a very relevant paper in the field that has now been added to our reference list.

Line 68 to line 82. Why is it avoided the most representative social groups of social production groups (ie laying hens, free range chicken) where there are groups only of females or only of males that although it is not a "natural" social group is the most frequent in management of these production systems?

Re: This study is a starting point for our assessment of social dynamics using these combined methodologies. Thus we started using a simple laboratory model with mixed sex groups. Other social groups that are most representative of poultry production systems are within our plans, and hopefully they will be carried out in the near future. The following modification was made in the Introduction (changes are underlines):

Experimentally, the social groups corresponded to triads of 2 females and 1 male. Although this ratio 2:1 (female: male) is not typically used in poultry commercial production systems, in our laboratory setup it allowed assessment of female-female as well as female-male interactions, while avoiding well documented violent male-male aggressions.

Line 103. Temporal organization of the group?

Re: The sentence was slightly modified for clarify (change is underlined)

We hypothesized that the presence of a dominant bird has an effect not only on the type of behaviors displayed by its subordinate but also on the temporal organization of the subordinate's behavior.

Line 134. Endpoint criteria provided in cases of extreme aggression?

Re: Yes, we did implement an endpoint criterion. This information has now been added to the Materials and Methods section as follows:

Birds were monitored remotely through the camera-computer setup at least on 3 time points throughout the day. Testing was interrupted if signs of physical injuries were apparent or considered that withstanding behavior could lead to physical injuries.

Noteworthy, the test originally was planned for a 5-day period, but was interrupted on the afternoon of the third day given the observed aggressions in particular within group 9.

Line 140-142 The definition of the group called neutral is not clear, it seems that these animals can eventually move to well-defined categories as dominant and subordinate?

Re: We considered relevant the categorization of neutral groups because neither aggressive dominantes, nor subordinates that suffer aggression were evident during the whole study. Moreover, living within neutral groups lead to the possibility of independent (not synchronized) behavior between the members of the group. Surely it is possible that those animals could move either to the dominant or the subordinate category in a future, but at least during the length of this study, that was not the case. Nevertheless, the definition of neutral has been rewritten as follows:

Five social groups were considered neutral given that all triad members showed aggressive scores between -4.9 and 4.9, thus clear hierarchal ranks were not evident during the study (Fig. 1f).

Line 355-365 How do you think that subordinate birds can be affected their productive efficiency? ... Was growth and conversion of food measured? ... Is it likely that these birds that spend more time walking develop a strategy to obtain resources with an avoidant personality of the dominant ones and which are finally more efficient? ... Can the energy cost of the dominant ones be higher than the energy cost of developing a strategy of avoidance and a better use of resources?

Re: These are very interesting questions and would have been very interesting to be able to look directly at this point. However, we found many difficulties for accurately measuring feed consumption in our experimental setup. First there were 3 birds and one tolva-like feeder thus we could not estimate individual feed consumption. Second, the feeders were designed to hold enough food to withstand a whole week, but feed spilling out some feed did occur, further defaulting estimation of feed consumption. We did weigh birds weekly up to 9 weeks of age, and did not detect any differences in weight between subordinate and dominate birds before the experiment. Because the experiment only lasted 3 days' significant weight loss during testing was not expected to be found given that it most likely would require more time to be detectable. In the future, with a longer experiment and improved setup this definitely would be an interesting study to make.

Line 366-374 Maybe it is more appropriate to talk about different "personalities" or adaptive strategies or coping styles of individuals and maybe it is good to review the literature in this regard, I recommend citing the following work:

Miranda-de la Lama, G. C., Pascual-Alonso, M., Aguayo-Ulloa, L., Sepúlveda, W. S., Villarroel, M., & María, G. A. (2019). Social personality in sheep: Can social strategies predict individual differences in cognitive abilities, morphology features and reproductive success?. *Journal of Veterinary Behavior*.

Re: We have now added a discussion regarding personalities/coping strategies as follows:

In our study, the majority of neutral groups were composed of individuals that were a priori selected based on low fearfulness and non-aggressiveness in behavioral tests (for details see Caliva et al., 2019). Thus, these neutral birds could present a specific coping style³⁹ /personality³⁹⁻⁴¹ which favor positive social interaction. Previous studies have shown that quail selected by their high adrenocortical response to restraint (i.e. proposed to have a reactive personality⁴²), are more fearful in a wide variety of tests⁴³ but also more aggressive in social groups⁴⁴, in comparison with those with low responsiveness (i.e. proactive personality⁴²). Also, quail selected as chicks as highly sociable, are less fearful and less aggressive in social groups as juveniles than less sociable birds¹⁸. In sheep, analysis of social behavior and the index of success of displacement has suggest the existence of at least 4 personality profiles (avoider, affiliative, aggressive, and pragmatic)⁴⁵. In their study, sheep with the avoider and affiliative profiles do not use aggressive behaviors, but rather the nonagonistic behaviors (i.e. licking, grooming, sniffing) as their predominant social strategy. It is possible that in our study all members of the neutral groups had profiles similar to the avoider and affiliative profiles, thus use nonagonistic behaviors as their predominant social strategy.

Page 18. Density and lighting are two of the aspects that affect the social groups of bird production. In your case, the density is not inconvenient and the lighting follows a natural regime. Taking into account the high densities (ie broiler) used in production and the 24-hour light rhythm during the first and last stage of fattening, the results obtained here can be useful in intensive bird production systems?

Re: Most likely in broiler raised for meat production the knowledge gained from this study is not expected to be replicated. This is due to the fact that in production systems, thousands of birds share the same space and are exposed to a light regime aimed at reducing aggression and maximizing growth. However, in other poultry setups, with lower density, such as broiler and layer breeders, or in the new layers systems with ground access, where all have more space to ambulate and to perform social interactions, our results could provide insight for the use of time series analysis for evaluation dominance/subordinate status and behavioral synchronization. Also, outside Europe there is still many places that hens and quail are still raised in cages in small groups and also in backyards small cage-like setups. In these contexts, the results of this study should also be useful.

Line 479. Explain the “broad implication for farm animal welfare”...

Re: The sentence has now been rewritten as follows: “These findings may have broad implications for farm and zoo animal welfare by providing a framework for analysis of behavioral synchronization between dominant/subordinate individuals, as well as pave the way for exploring strategies to counteract or help in controlling aggressive behaviors.”

Line 485. Give more information about the feeding regime of the animals and performance traits (consumption, efficiency, diet etc.)

Re: In Materials and Methods more information about the feeding regime has been added as follows:

Food and water were provided ad libitum. Both started and layer feeds were commercially obtained (20% of Crude Protein and 2900 kcal of Metabolizable Energy/kg diet). Feed contained corn, disabled soybean, wheat bran, soybean pellets, sunflower pellets, calcium, salt, vitamins, minerals and phosphate⁵⁹. Although in this study feed consumption was not assessed, previous studies in our laboratory under similar conditions show a weekly feed intake of adults of 212±2 g (~30 g daily)⁵⁹. Birds were weighed at 28 days of age, and the weight of birds transferred to cages ranged between 100-150g. Thereafter, weight was recorded weekly until 9 weeks of age and then at 92 days of age. At these same time points, male gonadal measurements showed complete development (Cloacal gonadal volume CGV>1000 mm³) in all males by 9 weeks of age. Female quail egg production was monitored throughout the study and all females reached peak egg production.

REVIEWERS' COMMENTS:

Reviewer #1 (Remarks to the Author):

The revised manuscript has fully addressed my questions and comments. The manuscript is now significantly improved, and I recommend publication.

Reviewer #2 (Remarks to the Author):

This study investigated the effects of social environment on birds' behavioural complexity. The authors managed to quantitatively evaluate changes in the animals' behavioural repertoire and, particularly, in the temporal dynamics of their behaviour as hierarchy established. They found out that subordinates performed less complex temporal dynamics of locomotion, and that their locomotor pattern was synchronised with locomotion pattern of dominants.

The authors paid attention to most aspects raised in my review. Follow, I list some suggestions of minor changes. After that, I recommend publication of the manuscript.

1 – In my previous review, I had recommended they informed on Table 1 the time unity used (seconds), since this information was missing. They included a sentence as a footnote of the Table; I however suggest them to include the information in the title of the Table, as below. "Table 1. Time (in seconds*) spent performing behaviors during 1-hour immediately after birds were placed in a novel social group (day 1), and 48 hours later (day 3)."

Since they informed one of the behaviours was exceptionally expressed as percentage of time on the Table, this specific information should be maintained in the footnote. E.g.: "*The behavior "ambulation" is exceptionally expressed as a percentage of time (%)."

2 – Line 172 – Please, exclude "a" and one comma from the sentence: "...these time series with lower α -value have higher fractal dimensions".

3 – In the previous review, I pointed to a result which had not been discussed in the Discussion section (Lines 253-255: "The social groups with a dominant individual showed a significantly higher ($P = 0.03$) mean interindividual distance between group members than the groups with only neutral members"). In this new version, the authors discussed the result, but without any citations to supporting studies. Therefore, I suggest the authors to include references to adequate bibliography, in order to support their statements/interpretations.

Reviewer #3 (Remarks to the Author):

The authors answer almost all my questions and I agree with them. I think that article is now suitable for publication in the Journal. The authors improve the original manuscript and they did a very good job.

Reviewer #2 (Remarks to the Author):

This study investigated the effects of social environment on birds' behavioural complexity. The authors managed to quantitatively evaluate changes in the animals' behavioural repertoire and, particularly, in the temporal dynamics of their behaviour as hierarchy established. They found out that subordinates performed less complex temporal dynamics of locomotion, and that their locomotor pattern was synchronised with locomotion pattern of dominants.

The authors paid attention to most aspects raised in my review. Follow, I list some suggestions of minor changes. After that, I recommend publication of the manuscript.

1 – In my previous review, I had recommended they informed on Table 1 the time unity used (seconds), since this information was missing. They included a sentence as a footnote of the Table; I however suggest them to include the information in the title of the Table, as below.

“Table 1. Time (in seconds*) spent performing behaviors during 1-hour immediately after birds were placed in a novel social group (day 1), and 48 hours later (day 3).”

Since they informed one of the behaviours was exceptionally expressed as percentage of time on the Table, this specific information should be maintained in the footnote.

E.g.: “*The behavior “ambulation” is exceptionally expressed as a percentage of time (%).”

Re: In Table 1 title and footnote have been modified accordingly.

2 – Line 172 – Please, exclude “a” and one comma from the sentence: “...these time series with lower α -value have higher fractal dimensions”.

Re: Text has been modified accordingly (see line 212).

3 – In the previous review, I pointed to a result which had not been discussed in the Discussion section (Lines 253-255: “The social groups with a dominant individual showed a significantly higher ($P = 0.03$) mean interindividual distance between group members than the groups with only neutral members”). In this new version, the authors discussed the result, but without any citations to supporting studies. Therefore, I suggest the authors to include references to adequate bibliography, in order to support their statements/interpretations.

Re: Appropriate citations have been added in lines 324-328, as follows (changes are underlined):

(...) Consistently, social proximity, which is commonly associated with underlying sociality (motivation to be near conspecifics) and social cohesion³⁹, was also different between both group types. Groups with a dominant individual show lower social proximity (larger interindividual distance), than the groups with only neutral members, which is consistent with shyness and social withdrawal of subordinates, and decreased social cohesion. Guzman et al.¹⁸ in quail classified as chicks as highly

sociable in a social proximity test (i.e. density-related permanence test) also observed lower average distance between birds and lower levels of aggression in comparison to those with low sociability¹⁸. Similarly, quail selected by their low andrenocortical response stay closer together as chicks³⁹ and show lower aggressiveness³⁹ as adults in comparison with those with high responsiveness. (...)